# Variational autoencoders in the presence of low-dimensional data: landscape and implicit bias

Frederic Koehler[* 1], Viraj Mehta[*2], Chenghui Zhou[*3], and Andrej Risteski[3]

[1]Department of Computer Science, Stanford University, `fkoehler@stanford.edu`
[2]Robotics Institute, Carnegie Mellon University, `virajm@cs.cmu.edu`
[3]Machine Learning Department, Carnegie Mellon University, {`chenghuz,`
`aristesk`}`@andrew.cmu.edu`

## Abstract

Variational Autoencoders (VAEs) are one of the most commonly used generative models, particularly for image data. A prominent difficulty in training VAEs is data that is supported on a lower dimensional manifold. Recent work by Dai and Wipf (2020) proposes a two-stage training algorithm for VAEs, based on a conjecture that in standard VAE training the generator will converge to a solution with 0 variance which is correctly supported on the ground truth manifold. They gave partial support for this conjecture by showing that some optima of the VAE loss do satisfy this property, but did not analyze the training dynamics. In this paper, we show that for linear encoders/decoders, the conjecture is true—that is the VAE training does recover a generator with support equal to the ground truth manifold—and does so due to an implicit bias of gradient descent rather than merely the VAE loss itself. In the nonlinear case, we show that VAE training frequently learns a higher-dimensional manifold which is a superset of the ground truth manifold.

## 1 Introduction

Variational autoencoders (VAEs) have recently enjoyed a revived interest, both due to architectural choices that have led to improvements in sample quality (Oord et al., 2017; Razavi et al., 2019b; Vahdat & Kautz, 2020) and due to algorithmic insights (Dai et al., 2017; Dai & Wipf, 2019). Nevertheless, fine-grained understanding of the behavior of VAEs is lacking, both on the theoretical and empirical level.

In our paper, we study a common setting of interest where the data is supported on a low-dimensional manifold — often argued to be the setting relevant to real-world image and text data due to the *manifold hypothesis* (see e.g. Goodfellow et al. (2016)). In this setting, Dai and Wipf (2019) proposed a two-stage training process for VAEs, based on a conjecture that for standard VAE training with such data distributions: (1) the generator's covariance will converge to 0, (2) the generator will learn the correct manifold, but not the correct density on the manifold (3) the number of approximately 0 eigenvalues in the encoder covariance will equal the intrinsic dimensionality of the manifold (see also Dai et al. (2017)). Formally, they showed that some optima of the VAE loss satisfy this conjecture, but they did not attempt to analyze the training dynamics.

In this paper, we revisit this setting and explore the behaviour of both the VAE loss, and the training dynamics. Through a combination of theory and experiments we show that:

- In the case of the data manifold being **linear** (i.e. the data is Gaussian, supported on a linear subspace—equivalently, it is produced as the pushforward of a Gaussian through a linear map), and the encoder/decoder being parametrized as linear maps, we show that: a) the set of optima includes parameters for which the generator's support is a *strict superset* of the data manifold; b)

---

[*]These authors contributed equally to this work.

the gradient descent dynamics are such that they converge to generators with support *equal* to the support of the data manifold. This provides a full proof of the conjecture in Dai & Wipf (2019), albeit we show the phenomenon is a combination of both the *location* of the minima of the loss as well as an *implicit bias of the training dynamics*.

- In the case of the data manifold being **nonlinear** (i.e. the data distribution is the pushforward of the Gaussian through a nonlinear map $f : \mathbb{R}^r \to \mathbb{R}^d, r \leq d$), the gradient descent dynamics from a random start often converges to generators $G$ whose support *strictly contains* the support of the underlying data distribution, while driving reconstruction error to 0 and driving the VAE loss to $-\infty$. This shows that the conjecture in Dai & Wipf (2019) does not hold for general nonlinear data manifolds and architectures for the generator/encoder.

**Organization:** We will provide an informal overview of our findings in Section 3. The rigorous discussion on the VAE landscape are in Section 4 and on the implicit bias of gradient descent in Section 5.

## 2 SETUP

We will study the behavior of VAE learning when data lies on a low-dimensional manifold—more precisely, we study when the generator can recover the support of the underlying data distribution. In order to have a well-defined "ground truth", both for our theoretical and empirical results, we will consider synthetic dataset that are generated by a "ground truth" generator as follows.

**Data distribution:** To generate a sample point $x$ for the data distribution, we will sample $z \sim N(0, I_{r^*})$, and output $x = f(z)$, for a suitably chosen $f$. In the *linear* case, $f(z) = Az$, for some matrix $A \in \mathbb{R}^{d \times r^*}$. In the *nonlinear* case, $f(z)$ will be a nonlinear function $f : \mathbb{R}^{r^*} \to \mathbb{R}^d$. We will consider several choices for $f$.

**Parameterization of the trained model:** For the model we are training, the generator will sample $z \sim N(0, I_r)$ and output $x \sim N(f(z), \epsilon^2 I)$, for trainable $f, \epsilon$; the encoder given input $x$ will output $z \sim N(g(x), D)$, where $D \in \mathbb{R}^{r \times r}$ is a diagonal matrix, and $g, D$ are trainable. In the linear case, $f, g$ will be parameterized as matrices $\tilde{A}, \tilde{B}$; in the nonlinear case, several different parameterizations will be considered. In either case the VAE Loss will be denoted by $L(\cdot)$, see (3).

## 3 OUR RESULTS

**Linear VAEs: the correct distribution is not recovered.** Recall in the linear case, we train a linear encoder and decoder to learn a Gaussian distribution consisting of data points $x \sim N(0, \Sigma)$ — equivalently, the data distribution is the pushforward of a standard Gaussian $z \sim N(0, I_r)$ through a linear generator $x = Az$ with $AA^T = \Sigma$; see also Section 2 above. In Theorem 1 of Lucas et al. (2019), the authors proved that in a certain probabilistic PCA setting where $\Sigma$ is full-rank, the landscape of the VAE loss has no spurious local minima: at any global minima of the loss, the VAE decoder exactly matches the ground truth distribution, i.e. $\tilde{A}\tilde{A}^T + \epsilon^2 I = \Sigma$.

We revisit this problem in the setting where $\Sigma$ has rank less than $d$ so that the data lies on the *lower-dimensional manifold/subspace* spanned by the columns of $A$ or equivalently $\Sigma$, denoted span$(A)$. We show empirically (i.e. via simulations) in Section 6 that when $\Sigma$ is rank-degenerate the VAE actually *fails to recover the correct distribution*. More precisely, the recovered $\tilde{A}$ has the correct column span but fails to recover the correct density — confirming predictions made in Dai & Wipf (2019). We then explain theoretically why this happens, where it turns out we find some surprises.

**Landscape Analysis: Linear and Nonlinear VAE.** Dai & Wipf (2019) made their predictions on the basis of the following observation about the *loss landscape*: there can exist sequences of VAE solutions whose objective value approaches $-\infty$ (i.e. are asymptotic global minima), for which the generator has the correct column span, but does not recover the correct density on the subspace. They also informally argued that these are all of the asymptotic global minima of loss landscape (Pg 7 and Appendix I in Dai & Wipf (2019)), but did not give a formal theorem or proof of this claim.

We settle the question by showing this is *not the case*:[*] namely, there exist many convergent sequences of VAE solutions which still go to objective value $-\infty$ but also do not recover the correct

---

[*] They also argued this would hold in the nonlinear case, but our simulations show this is generally false in that setting, even for the solutions chosen by gradient descent with a standard initialization — see Section 6.

column span — instead, the span of such $\tilde{A}$ is a strictly larger subspace. More precisely, we obtain a tight characterization of all asymptotic global minima of the loss landscape:

**Theorem 1** (Optima of Linear VAE Loss, Informal Version of Theorem 3). *Suppose that $\tilde{A}, \tilde{B}$ are fixed matrices such that $A = \tilde{A}\tilde{B}A$ and suppose that $\#\{i : \tilde{A}_i = 0\} > r - d$, i.e. the number of zero columns of $\tilde{A}$ is strictly larger than $r - d$. Then there exists $\tilde{\epsilon}_t \to 0$ and positive diagonal matrices $\tilde{D}_t$ such that $\lim_{t\to\infty} L(\tilde{A}, \tilde{B}, \tilde{D}_t, \tilde{\epsilon}_t) = -\infty$. Also, these are all of the asymptotic global minima: any convergent sequence of points $(\tilde{A}_t, \tilde{B}_t, \tilde{D}_t, \tilde{\epsilon}_t)$ along which the loss $L$ goes to $-\infty$ satisfies $\tilde{A}_t \to \tilde{A}, \tilde{B}_t \to \tilde{B}$ with $A = \tilde{A}\tilde{B}A$ such that $\#\{i : \tilde{A}_i = 0\} > r - d$.*

To interpret the constraint $\#\{i : \tilde{A}_i = 0\} > r - d$, observe that if the data lies on a lower-dimensional subspace of dimension $r^* < d$ (i.e. $r^*$ is the rank of $\Sigma$), then there exists a generator which generates the distribution with $r - r^* > r - d$ zero columns by taking an arbitrary low-rank factorization $LL^T = \Sigma$ with $L : d \times r^*$ and defining $A : d \times r$ by $A = [L \quad 0_{d \times r - r^*}]$. The larger the gap is between the manifold/intrinsic dimension $r^*$ and the ambient dimension $d$, the more flexibility we have in constructing asymptotic global minima of the landscape. Also, we note there is no constraint in the Theorem that $r - d \geq 0$: the assumption is automatically satisfied if $r < d$.

To summarize, the asymptotic global minima satisfy $A = \tilde{A}\tilde{B}A$, so the column span of $\tilde{A}$ contains that of $A$, but in general it can be a higher dimensional space. For example, if $d, r \geq r^* + 2$ and and the ground truth generator is $A = \begin{bmatrix} I_{r^*} & 0 \\ 0 & 0 \end{bmatrix}$, then $\tilde{A} = \begin{bmatrix} I_{r^*+1} & 0 \\ 0 & 0 \end{bmatrix}$ and $\tilde{B} = \begin{bmatrix} I_{r^*+1} & 0 \\ 0 & 0 \end{bmatrix}$ is a perfectly valid asymptotic global optima of the landscape, even though decoder $\tilde{A}$ generates a different higher-dimensional Gaussian distribution $N\left(0, \begin{bmatrix} I_{r^*+1} & 0 \\ 0 & 0 \end{bmatrix}\right)$ than the ground truth. In the above result we showed that there are asymptotic global minima with higher dimensional spans even with the common restriction that the encoder variance is diagonal; if we considered the case where the encoder variance is unconstrained, as done in Dai & Wipf (2019), and/or can depend on its input $x$, this can only increase the number of ways to drive the objective value to $-\infty$.

We also consider the analogous question in the *nonlinear* VAE setting where the data lies on a low-dimensional manifold. We prove in Theorem 6 that even in a very simple example where we fit a VAE to generate data produced by a 1-layer ground truth generator, there exists a bad solution with strictly larger manifold dimension which drives the reconstruction error to zero (and VAE loss to $-\infty$). The proof of this result does not depend strongly on the details of this setup and it can be adapted to construct bad solutions for other nonlinear VAE settings.

We note that the nature both of these result is asymptotic: that is, they consider sequences of solutions whose loss converges to $-\infty$ — but not the rate at which they do so. In the next section, we will consider the trajectories the optimization algorithm takes, when the loss is minimized through gradient descent.

**Linear VAE: implicit regularization of gradient flow.** The above theorem indicates that studying the minima of the loss landscape alone cannot explain the empirical phenomenon of linear VAE training recovering the support of the ground truth manifold in experiments; the only prediction that can be made is that the VAE will recover a *possibly larger* manifold containing the data.

We resolve this tension by proving that *gradient flow*, the continuous time version of gradient descent, has an *implicit bias* towards the low-rank global optima. Precisely, we measure the effective rank quantitatively in terms of the singular values: namely, if $\sigma_k(\tilde{A})$ denotes the $k$-th largest singular value of matrix $\tilde{A}$, we show that all but the largest $\dim(\text{span } A)$ singular values of $\tilde{A}$ decay at an exponential rate, as long as: (1) the gradient flow continues to exist[*] , and (2) the gradient flow does not go off to infinity, i.e. neither $\tilde{A}$ or $\tilde{\epsilon}$ go to infinity (in simulations, the decoder $\tilde{A}$ converges to a bounded point and $\tilde{\epsilon} \to 0$ so the latter assumption is true). To simplify the proof, we work with a slightly modified loss which "eliminates" the encoder variance by setting it to its optimal value:

---

[*]We remind the reader that the gradient flow on loss $L(x)$ is a differential equation $dx/dt = -\nabla L(x)$. Unlike discrete-time gradient descent, gradient flow in some cases (e.g. $dx/dt = x^2$) has solutions which exist only for a finite time (e.g. $x = 1/(1 - t)$), which "blows up" at $t = 1$), so we need to explicitly assume the solution exists up to time $T$.

$L_1(\tilde{A}, \tilde{B}, \tilde{\epsilon}) := \min_{\tilde{D}} L(\tilde{A}, \tilde{B}, \tilde{\epsilon}, \tilde{D})$; this loss has a simpler closed form, and we believe the theorems should hold for the standard loss as well. (Generally, gradient descent on the original loss $L$ will try to optimize $\tilde{D}$ in terms of the other parameters, and if it succeeds to keep $\tilde{D}$ well-tuned in terms of $\tilde{A}, \tilde{B}, \tilde{\epsilon}$ then $L$ will behave like the simplified loss $L_1$.)

**Theorem 2** (Implicit Bias of Gradient Flow, Informal version of Theorem 5). *Let $A : d \times r$ be arbitrary and define $W$ to be the span of the rows of $A$, let $\tilde{\Theta}(0) = (\tilde{A}(0), \tilde{B}(0), \tilde{\epsilon}(0))$ be an arbitrary initialization and define the gradient flow $\tilde{\Theta}(t) = (\tilde{A}(t), \tilde{B}(t), \tilde{\epsilon}(t))$ by the ordinary differential equation (ODE)*

$$\frac{d\tilde{\Theta}(t)}{dt} = -\nabla L_1(\tilde{\Theta}(t)) \tag{1}$$

*with initial condition $\Theta_0$. If the solution to this equation exists on the time interval $[0, T]$ and satisfies $\max_{t \in [0,T]} \max_j [\|(\tilde{A}_t)_j\|^2 + \tilde{\epsilon}_t^2] \leq K$, then for all $t \in [0, T]$ we have*

$$\sum_{k=\dim(W)+1}^{d} \sigma_k^2(\tilde{A}(t)) \leq C(A, \tilde{A})\, e^{-t/K} \tag{2}$$

*where $C(A, \tilde{A}) := \|P_{W^\perp} \tilde{A}^T(0)\|_F^2$ and $P_{W^\perp}$ is the orthogonal projection onto the orthogonal complement of $W$.*

Together, our Theorem 1 and Theorem 2 show that if gradient descent converges to a point while driving the loss to $-\infty$, then it successfully recovers the ground truth subspace/manifold span $A$. This shows that, in the linear case, the conjecture of Dai & Wipf (2019) can indeed be validated provided we incorporate training dynamics into the picture. The prediction of theirs we do not prove is that the number of zero entries of the encoder covariance $D$ converges to the intrinsic dimension; as shown in Table 1, in a few experimental runs this does not occur — in contrast, Theorem 2 implies that $\tilde{A}$ should have the right number of nonzero singular values and our experiments agree with this.

**Nonlinear VAE: Dynamics and Simulations.** In the linear case, we showed that the implicit bias of gradient descent leads the VAE training to converge to a distribution with the correct support. In the nonlinear case, we show that this does not happen—even in simple cases.

Precisely, in the setup of the one-layer ground truth generator, where we proved (Theorem 6) there exist bad global optima of the landscape, we verify experimentally (see Figure 1) that gradient descent from a random start does indeed converge to such bad asymptotic minima. In particular, this shows that whether or not gradient descent has a favorable implicit bias strongly depends on the data distribution and VAE architecture.

More generally, by performing experiments with synthetic data of known manifold dimension, we make the following conclusions: (1) gradient descent training recovers manifolds approximately containing the data, (2) these manifolds are generally not the same dimension as the ground truth manifold, but larger (this is in contrast to the conjecture in Dai & Wipf (2019) that they should be equal) even when the decoder and encoder are large enough to represent the ground truth and the reconstruction error is driven to 0 (VAE loss is driven to $-\infty$), and (3) of all manifolds containing the data, gradient descent still seems to favor those with relatively low (but not always minimal) dimension. Further investigating the precise role of VAE architecture and optimization algorithm, as well as the interplay with the data distribution is an exciting direction for future work.

## 3.1 RELATED WORK

**Implicit regularization.** Interestingly, the implicit bias towards low-rank solutions in the VAE which we discover is consistent with theoretical and experimental results in other settings, such as deep linear networks/matrix factorization (e.g. Gunasekar et al. (2018); Li et al. (2018); Arora et al. (2019); Li et al. (2020); Jacot et al. (2021)), although it seems to be for a different mathematical reason — unlike those settings, initialization scale does not play a major role. Similar to the setting of implicit margin maximization (see e.g. Ji & Telgarsky (2018); Schapire & Freund (2013); Soudry et al. (2018)), in our VAE setting the optima are asymptotic (though approaching a finite point, not off at infinity) and the loss goes to $-\infty$. Kumar & Poole (2020); Tang & Yang (2021) also explore some implicit regularization effects tied to the Jacobian of the generator and the covariance of the Gaussian noise.

**Architectural and Algorithmic Advances for VAEs.** There has been a recent surge in activity with the goal of understanding VAE training and improving its performance in practice. Much of the work has been motivated by improving posterior modeling to avoid problems such as "posterior collapse", see e.g. (Dai et al., 2020; Razavi et al., 2019a; Pervez & Gavves, 2021; Lucas et al., 2019; He et al., 2019; Oord et al., 2017; Razavi et al., 2019b; Vahdat & Kautz, 2020). Most relevant to the current work are probably the works Dai & Wipf (2019) and Lucas et al. (2019) discussed earlier. A relevant previous work to these is Dai et al. (2017); one connection to the current work is that they also performed experiments with a ground truth manifold, in their case given as the pushforward of a Gaussian through a ReLU network. In their case, they found that for a certain decoder and encoder architectures that they could recover the intrinsic dimension using a heuristic related to the encoder covariance eigenvalues from Dai & Wipf (2019); our results are complementary in that they show that this phenomena is not universal and does not hold for other natural datasets (e.g. manifold data on a sphere fit with a standard VAE architecture).

## 4 VAE LANDSCAPE ANALYSIS

In this section, we analyze the landscape of a VAE, both in the linear and non-linear case.

**Preliminaries and notation.** We use a VAE to model a datapoint $x \in \mathbb{R}^d$ as the pushforward of $z \sim N(0, I_r)$. We have the following standard VAE architecture:

$$p(x|z) = N(f(z), \epsilon^2 I), \qquad q(z|x) = N(g(x), D)$$

where $\epsilon^2 > 0$ is the decoder variance, $D$ is a diagonal matrix with nonnegative entries, and $f, g, D, \epsilon$ are all trainable parameters. (For simplicity, our $D$ does not depend on $x$; this is the most common setup in the linear VAE case we will primarily focus on.) The VAE objective (see Appendix A for explicit derivation) is to *minimize*:

$$L(f, g, D, \epsilon) := \mathbb{E}_{x \sim p^*} \mathbb{E}_{z' \sim N(0, I_r)} \left[ \frac{1}{2\epsilon^2} \|x - f(g(x) + D^{1/2}z')\|^2 + \|g(x)\|^2/2 \right]$$
$$+ d \log(\epsilon) + \operatorname{Tr}(D)/2 - \frac{1}{2} \sum_i \log D_{ii}. \tag{3}$$

We also state a general fact about VAEs for the case that the objective value can be driven to $-\infty$, which was observed in (Dai & Wipf, 2019): they must satisfy $\epsilon \to 0$ and achieve perfect limiting reconstruction error. The first claim in this Lemma is established in the proof of Theorem 4 and the second claim is Theorem 5 in Dai & Wipf (2019). For completeness, we include a self-contained proof in Appendix B.1.

**Lemma 1** (Theorems 4 and 5 of Dai & Wipf (2019)). *Suppose $f_t, g_t, D_t, \epsilon_t$ for $t \geq 1$ are a sequence such that $\lim_{t \to \infty} L(f_t, g_t, D_t, \epsilon_t) = -\infty$. Then: 1) $\lim_{t \to \infty} \epsilon_t = 0$ and 2) $\lim_{t \to \infty} \mathbb{E}_{x \sim p^*} \mathbb{E}_{z' \sim N(0, I_r)} \|x - f_t(g_t(x) + D_t^{1/2}z')\|^2 = 0$.*

In fact, the reconstruction error and $\epsilon$ are closely linked in a simple way:

**Lemma 2.** *If $f, g, D$ are fixed, then the optimal value of $\epsilon$ to minimize $L$ is given by $\epsilon = \sqrt{\frac{1}{d} \mathbb{E}_{x \sim p^*} \mathbb{E}_{z' \sim N(0, I_r)} \left[ \|x - f(g(x) + D^{1/2}z')\|^2 \right]}$.*

### 4.1 LINEAR VAE

**Setup:** In the linear VAE case, we assume the data is generated from the model $x = Az$ with $A \in \mathbb{R}^{d \times r^*}$ and $z \sim \mathcal{N}(0, I_{r^*})$. We will denote the training parameters by $\tilde{A} \in \mathbb{R}^{d \times r}, \tilde{B} \in \mathbb{R}^{r \times d}$, $\tilde{D} \in \mathbb{R}^{r \times r}$, and $\tilde{\epsilon} > 0$, where $r \geq 1$ is a fixed hyperparameter which corresponds to the latent dimension in the trained generator, and we assume $\tilde{D}$ is a diagonal matrix. With this notation in place, the implied VAE has generator/decoder $\tilde{x} \sim \mathcal{N}(\tilde{A}z, \tilde{\epsilon}^2 I_d)$ and encoder $\tilde{z} \sim \mathcal{N}(\tilde{B}x, \tilde{D})$. The VAE objective as a function of parameters $\tilde{\Theta} = (\tilde{A}, \tilde{B}, \tilde{D}, \tilde{\epsilon})$ is (see Appendix A):

$$L(\tilde{\Theta}) = \frac{1}{2\tilde{\epsilon}^2} \|A - \tilde{A}\tilde{B}A\|_F^2 + \frac{1}{2} \|\tilde{B}A\|_F^2 + d \log \tilde{\epsilon} + \frac{1}{2} \sum_i \left( \tilde{D}_{ii} \|\tilde{A}_i\|^2 / \tilde{\epsilon}^2 + \tilde{D}_{ii} - \log \tilde{D}_{ii} \right) \tag{4}$$

Our analysis makes use of a simplified objective $L_1$, which "eliminates" $D$ out of the objective by plugging in the optimal value of $D$ for a choice of the other variables. We use this as a technical tool when analyzing the landscape of the original loss $L$.

**Lemma 3** (Deriving the simplified loss $L_1$). *Suppose that $\tilde{A}, \tilde{B}, \tilde{\epsilon}$ are fixed. Then the objective $L$ is minimized by choosing for all $i$ that $\tilde{D}_{ii} = \frac{\tilde{\epsilon}^2}{\|\tilde{A}_i\|^2 + \tilde{\epsilon}^2}$ where $\tilde{A}_i$ is column $i$ of $\tilde{A}$, and for $L_1(\tilde{A}, \tilde{B}, \tilde{\epsilon}) := \min_{\tilde{D}} L(\tilde{A}, \tilde{B}, \tilde{D}, \tilde{\epsilon})$ it holds that*

$$L_1(\tilde{A}, \tilde{B}, \tilde{\epsilon}) = \frac{1}{2\tilde{\epsilon}^2}\|A - \tilde{A}\tilde{B}A\|_F^2 + \frac{1}{2}\|\tilde{B}A\|_F^2 + (d-r)\log\tilde{\epsilon} + \sum_i \frac{1 + \log\left(\|\tilde{A}_i\|^2 + \tilde{\epsilon}^2\right)}{2}. \quad (5)$$

Taking advantage of this simplified formula, we can then identify (for the original loss $L$) simple sufficient conditions on $\tilde{A}, \tilde{B}$ which ensure they can be used to approach the population loss minimum by picking suitable $\tilde{\epsilon}_t, \tilde{D}_t$ and prove matching necessary conditions.

**Theorem 3.** *First, suppose that $\tilde{A} : d \times r, \tilde{B} : r \times d$ are fixed matrices such that $A = \tilde{A}\tilde{B}A$ and suppose that $\#\{i : \tilde{A}_i = 0\} > r - d$, i.e. the number of zero columns of $\tilde{A}$ is strictly larger than $r - d$. Then for any sequence of positive $\tilde{\epsilon}_t \to 0$ there exist a sequence of positive diagonal matrices $\tilde{D}_t$ such that:*

1. *For every $i$ such that $\tilde{A}_i \neq 0$, i.e. column $i$ of $\tilde{A}$ is nonzero, we have $(\tilde{D}_t)_{ii} \to 0$.*

2. *$\lim_{t\to\infty} L(\tilde{A}, \tilde{B}, \tilde{D}_t, \tilde{\epsilon}_t) = -\infty$.*

*Conversely, suppose that that $\tilde{A}_t, \tilde{B}_t, \tilde{D}_t, \tilde{\epsilon}_t$ is an arbitrary sequence such that $\lim_{t\to\infty} L(\tilde{A}_t, \tilde{B}_t, \tilde{D}_t, \tilde{\epsilon}_t) = -\infty$. Then as $t \to \infty$, we must have that:*

1. *$\tilde{\epsilon}_t \to 0$ and $\|A - \tilde{A}_t\tilde{B}_tA\|_F^2 \to 0$.*

2. *$\max_i(\tilde{D}_t)_{ii}\|(\tilde{A}_t)_i\|_F^2 \to 0$ where $(\tilde{A}_t)_i$ denotes the $i$-th column of $\tilde{A}_t$.*

3. *For any $\delta > 0$, $\liminf_{t\to\infty} \#\{i : \|(\tilde{A}_t)_i\|_2^2 < \delta\} > r - d$, i.e. asymptotically $\tilde{A}_t$ has strictly more than $r - d$ columns arbitrarily close to zero.*

*In particular, if $(\tilde{A}_t, \tilde{B}_t, \tilde{D}_t, \tilde{\epsilon}_t)$ converge to a point $(\tilde{A}, \tilde{B}, \tilde{D}, \tilde{\epsilon})$ then $\tilde{\epsilon} = 0$, $A = \tilde{A}\tilde{B}A$, $\tilde{D}_{ii} = 0$ for every $i$ such that $\tilde{A}_i \neq 0$, and $\#\{i : \tilde{A}_i = 0\} > r - d$.*

For the sufficiency direction, we observe that in (5) the first term is zero if $A = \tilde{A}\tilde{B}A$ and the sum of the last two terms goes to $-\infty$ if $\tilde{\epsilon} \to 0$ and enough columns of $\tilde{A}$ are zero. Based upon similar reasoning combined with Lemma 1, we show necessity. The full proof is in the Appendix.

## 4.2 Nonlinear VAE

In this section, we give a simple example of a nonlinear VAE architecture which can represent the ground truth distribution perfectly, but has another asymptotic global minimum where it outputs data lying on a manifold of a larger dimension ($r^* + s$ instead of $r^*$ for any $s \geq 1$). The ground truth model is a one-layer network ("sigmoid dataset" in Section 6) and the bad decoder we construct outputs a standard Gaussian in $r^* + s$ dimensions padded with zeros.

**Theorem 4** (Theorem 6 in Appendix). *Let $s \geq 1$ be arbitrary and consider the sigmoid setup from Section 6. There exists $\tilde{A}_1, \tilde{A}_2, \tilde{B}$ s.t. for $\tilde{\epsilon}_t \to 0$ there exists $\tilde{D}_t$ s.t. (1) the VAE loss $L(\tilde{A}_1, \tilde{A}_2, \tilde{B}, \tilde{D}_t, \tilde{\epsilon}_t) \to -\infty$; (2) The output of the decoder $\tilde{x} = \tilde{A}_1\tilde{z} + \sigma(\tilde{A}_2\tilde{z})$, $\tilde{z} \sim N(0, I_r)$ is a standard Gaussian in the first $r^* + s$ coordinates and zero in the remaining ones.*

Thus, the generator constructed has as support a manifold of larger dimension ($r^* + s$). Moreover, in Section 6, we show that this is not merely a theoretical possibility: we show through simulations that gradient descent from a random initialization often converges to similar minima.

## 5 Implicit bias of gradient descent in Linear VAE

In this section, we prove that even though the landscape of the VAE loss contains generators with strictly larger support than the ground truth, the gradient flow is *implicitly biased towards low-rank solutions*. We prove this for the simplified loss $L_1(\tilde{A}, \tilde{B}, \tilde{\epsilon}) = \min_{\tilde{D}} L(\tilde{A}, \tilde{B}, \tilde{\epsilon}, \tilde{D})$, which makes

the calculations more tractable, though we believe our results should hold for the original loss $L$ as well. The main result we prove is as follows:

**Theorem 5** (Implicit bias of gradient descent). *Let $A : d \times r$ be arbitrary and define $W$ to be the span of the rows of $A$, let $\tilde{\Theta}(0) = (\tilde{A}(0), \tilde{B}(0), \tilde{\epsilon}(0))$ be an arbitrary initialization and define the gradient flow $\tilde{\Theta}(t) = (\tilde{A}(t), \tilde{B}(t), \tilde{\epsilon}(t))$ by the differential equation* (1). *with initial condition $\tilde{\Theta}_0$. If the solution to this equation exists on the time interval $[0, T]$ and satisfies $\max_{t \in [0,T]} \max_j [\|(\tilde{A}_t)_j\|^2 + \tilde{\epsilon}_t^2] \leq K$, then for all $t \in [0, T]$ we have*

$$\sum_{k=\dim(W)+1}^{d} \sigma_k^2(\tilde{A}(t)) \leq \|P_{W^\perp} \tilde{A}^T(t)\|_F^2 \leq e^{-t/K} \|P_{W^\perp} \tilde{A}^T(0)\|_F^2 \qquad (6)$$

*where $P_{W^\perp}$ is the orthogonal projection onto the orthogonal complement of $W$.*

Towards showing the above result, we first show how to reduce to matrices where $A$ has $d - \dim(\text{rowspan}(A))$ rows that are all-zero. To do this, we observe that the linear VAE objective is invariant to arbitrary rotations in the output space (i.e. $x$-space), so the gradient descent/flow trajectories transform naturally under rotations. Thus, we can "rotate" the ground truth parameters as well as the training parameters.

This is formally captured as Lemma 6 in the Appendix. Recall that by the singular value decomposition $A = USV^T$ for some orthogonal matrices $U, V$ and diagonal matrix $S$, and rotation invariance in the $x$-space lets us reduce to analyzing the case where $U = I$, i.e. $A = SV^T$. This matrix has a zero row for every zero singular value.

**Analysis when $A$ has zero rows.** Having reduced our analysis to the case where $A$ has zero rows, the following key lemma shows that for every $i$ such that row $i$ of $A$ (denoted $A^{(i)}$) is zero, the gradient descent step $-\nabla L$ or $-\nabla L_1$ will be negatively correlated with the corresponding row $\tilde{A}^{(i)}$.

**Lemma 4** (Gradient correlation). *If row $i$ of $A$ is zero, then*

$$\sum_{j=1}^{r} \tilde{A}_{ij} \frac{\partial L}{\partial \tilde{A}_{ij}} \geq \sum_{j=1}^{r} \tilde{D}_{jj} \tilde{A}_{ij}^2 / \tilde{\epsilon}^2, \qquad \sum_{j=1}^{r} \tilde{A}_{ij} \frac{\partial L_1}{\partial \tilde{A}_{ij}} \geq \sum_{j=1}^{r} \frac{\tilde{A}_{ij}^2}{\|\tilde{A}_j\|^2 + \tilde{\epsilon}^2}.$$

The way we use it is to notice that since the negative gradient points towards zero, gradient descent will shrink the size of $\tilde{A}^{(i)}$. Since the size of the matrix $\tilde{A}$ stays bounded, this should mean that for small step sizes the norm of row $i$ of $\tilde{A}$ shrinks by a constant factor at every step of gradient descent on loss $L_1$. We formalize this in continuous time for the gradient flow, i.e. the limit of gradient descent as step size goes to zero: for the special case of Theorem 2 in the zero row setting, the corresponding rows of $\tilde{A}$ decay exponentially fast.

**Lemma 5** (Exponential decay of extra rows). *Let $A$ be arbitrary, and let $\tilde{\Theta}(0) = (\tilde{A}(0), \tilde{B}(0), \tilde{\epsilon}(0))$ be an arbitrary initialization and define the gradient flow $\tilde{\Theta}(t) = (\tilde{A}(t), \tilde{B}(t), \tilde{\epsilon}(t))$ to be a solution of the differential equation* (1) *with initial condition $\tilde{\Theta}(0)$. If the solution exists on the time interval $[0, T]$ and satisfies $\max_{t \in [0,T]} \max_j [\|(\tilde{A}(t))_j\|^2 + \tilde{\epsilon}(t)^2] \leq K$ for some $K > 0$, then for all $i$ such that row $i$ of $A$ is zero we have $\|\tilde{A}^{(i)}(t)\|^2 \leq e^{-t/K} \|\tilde{A}^{(i)}(0)\|^2$ for all $t \in [0, T]$.*

## 6 SIMULATIONS

In this section, we provide extensive empirical support for the questions we addressed theoretically. In particular we investigate the kinds of minima VAEs converge to when optimized via gradient descent over the course of training.

**Linear VAEs:** First, we investigate whether linear VAEs are able to find the correct support for a distribution supported over a linear subspace. The setup is as follows. We choose a ground truth linear transformation matrix $A$ by concatenating an $r^* \times r^*$ matrix consisting of iid standard Gaussian entries with a zero matrix of dimension $(d - r^*) \times r^*$; the data is generated as $Az$, $z \sim \mathcal{N}(0, I^{r^*})$. Thus the data lies in a $r^*$-dimensional subspace embedded in a $d$-dimensional space. We ran the experiment with various choices for $r^*, d$ as well as the latent dimension of the trained decoder (Table 1). Every result is the mean over three experiments run with the same dimensionality and setup but a different random seed.

| Intrinsic Dimension | 3 | 3 | 6 | 6 | 9 | 9 | 12 |
|---|---|---|---|---|---|---|---|
| Ambient Dimension | 12 | 20 | 12 | 20 | 12 | 20 | 20 |
| Mean #0's in Encoder Variance | 3.3 | 3.7 | 6 | 6 | 9.3 | 9 | 12 |
| Mean # Decoder Rows Nonzero | 3 | 3 | 6 | 6 | 9 | 9 | 12 |
| Mean Normalized Eigenvalue Error | 0.44 | 0.71 | 0.49 | 0.47 | 0.30 | 0.45 | 0.42 |

Table 1: Optima found by training a linear VAE on data generated by a linear generator (i.e. a linearly transformed standard multivariate gaussian embedded in a larger ambient dimension by padding with zeroes) via gradient descent. The results reflect the predictions of Theorem 5: the number of nonzero rows of the decoder always match the dimensionality of the input data distribution with no variance while the number of nonzero dimensions of encoder variance is greater than or equal to the nonzero rows. All VAEs are trained with a 20-dimensional latent space. Clearly, the model fails to recover the correct eigenvalues and therefore has a substantially wrong data density function.

*Results:* From Table 1 we can see that the optima found by gradient descent capture the support of the manifold accurately across all choices of $d, r$, with the correct number of nonzero decoder rows. We also almost always see the correct number of zero dimensions in the diagonal matrix corresponding to the encoder variance.

However, gradient descent is unable to recover the density of the data on the learned manifold in the linear setting — in sharp contrast to the full rank case (Lucas et al., 2019). We conclude this by comparing the eigenvalues of the data covariance matrix and the learned generator covariance matrix. In order to understand whether the distribution on the linear subspace has the right density, we compute the eigenvalue error by forming matrices $X, \hat{X}$ with $n$ rows, for which each row is sampled from the ground truth and learned generator distribution respectively. We then compute the vector of eigenvalues $\lambda, \hat{\lambda}$ for the ground truth covariance matrix $AA^T$ and empirical covariance matrix $(1/n)\hat{X}^T\hat{X}$ respectively and compute the normalized eigenvalue error $||\hat{\lambda} - \lambda||/||\lambda||$. In no case does the density of the learned distribution come close to the ground truth.

**Nonlinear Dataset** In this section, we investigate whether VAEs are able to find the correct support in nonlinear settings. Unlike the linear setting, there is no "canonical" data distribution suited for a nonlinear VAE, so we explore two setups:

- *Sphere dataset:* The data are generated from the unit sphere concatenated with zero padding at the end. This can be interpreted as a unit sphere embedded in a higher dimensional space. We used 3 layers of 200 hidden units to parameterize our encoder and decoder networks.

  To measure how well the VAE has learnt the support of the distribution, we evaluate the average of $(||\tilde{x}_{:(r+1)}||_2 - 1)^2$, where $\tilde{x}$ are generated by the learnt generator. We will call this quantity *manifold error*. We have also evaluated the *padding error*, which is defined as $||\tilde{x}_{r+2:}||_2^2$.

- *Sigmoid Dataset:* Let $z \sim \mathcal{N}(0, I_r)$, the sigmoid dataset concatenates $z$ with $\sigma(\langle a^*, z \rangle)$ where $a^* \in \mathbb{R}^r$ is generated according to $\mathcal{N}(0, I_r)$. We add additional zero paddings to embed the generated data in a higher dimensional ambient space. The decoder is parameterized by a nonlinear function $f(z) = \tilde{A}z + \sigma(\tilde{C}z)$ and the encoder is parameterized by a linear function $g(x) = \tilde{B}x$. The intrinsic dimension of the dataset is $r$.

  To measure how well the VAE has learnt the support of the distribution, we evaluate the average of $(\sigma(\langle a^*, \tilde{x}_{:r} \rangle) - \tilde{x}_{r+1})^2$, where $\tilde{x}$ are generated by the learnt decoder. We will call this quantity *manifold error*. The *padding error* is defined as similarly as the sphere dataset.

*Results:* Due to space constraints, results for the sphere dataset (Table 3 and Figure 3) are in Appendix D. In both of the nonlinear dataset experiments, we see that the number of zero entries in the diagonal encoder variance is less reflective of the intrinsic dimension of the manifold than the linear dataset. It is, however, at least as large as the intrinsic dimension (Table 3, 2). We consider a coordinate to be 0 if it's less than 0.1. We found that the magnitude of each coordinate to be well separated, i.e. the smaller coordinates tend to be smaller than 0.1 and the larger tend to be bigger than 0.5. Thus the threshold selection is not crucial. We did not include padding error in the tables because it reaches zero in all experiments

We show the progression of manifold error, decoder variance and VAE loss during training for the sphere data in Figure 3 and for the sigmoid data in Figure 2, which are included in Appendix D due to space constraints. Datasets of all configurations of dimensions reached close to zero decoder

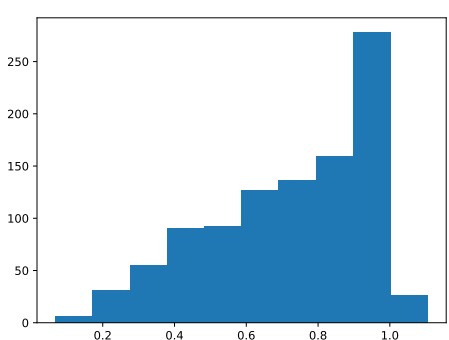 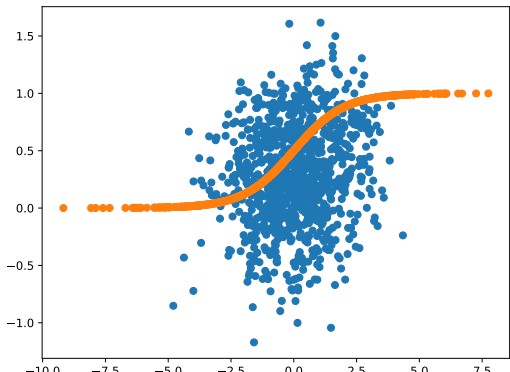

Figure 1: A demonstration that in the nonlinear setting (both types of data padded with zeroes to embed in higher ambient dimension, see Setup in Section 6) VAE training does not always recover a distribution with the correct support. *Left figure:* A histogram of the norms of samples generated from the VAE restricted to the dimensions which are not zero, which shows many of the points have norm less than 1. (The ground-truth distribution would output only samples of norm 1.) The particular example here is Column 2 in Table 3. *Right figure:* Two-dimensional linear projection of data output by VAE generator trained on our sigmoid dataset. The $x$-axis denotes $\langle a^*, \tilde{x}_{:r} \rangle$ and the $y$-axis is $\tilde{x}_{r+1}$, the blue points are from the trained VAE and the orange points are from the ground truth. In contrast to the ground truth data, which satisfies the sigmoidal constraint $x_{r+1} = \sigma(\langle a^*, x_{:r} \rangle)$, the trained VAE points do not and instead resemble a standard gaussian distribution. This is a case that closely resembles the example provided in Theorem 6. Also similar to Theorem 6, the VAE model plotted here (from Column 6 in Table 2) achieves nearly-perfect reconstruction error, approximately $0.001$.

| Intrinsic Dimensions | 3 | 3 | 5 | 5 | 7 | 7 |
|---|---|---|---|---|---|---|
| Ambient Dimensions | 7 | 17 | 11 | 22 | 15 | 28 |
| VAE Latent Dimensions | 6 | 8 | 10 | 16 | 13 | 24 |
| Mean Manifold Error | 0.09 | 0.13 | 0.23 | 0.24 | 0.18 | 0.28 |
| Mean #0's in Encoder Variance | 3 | 3.6 | 6 | 6.3 | 7.3 | 8 |

Table 2: Optima found by training a VAE on the sigmoid dataset. The VAE training consistently yields encoder variances with number of 0 entries greater than or equal to the intrinsic dimension.

variances, meaning the VAE loss is approaching $-\infty$. To demonstrate Theorem 6, we took examples from both datasets to visualize their output.

For the sphere dataset, we visualize the data generated from the model, with 8 latent dimensions, trained on unit sphere with 2 intrinsic dimensions and 16 ambient dimensions (Column 2 in Table 3). Its training progression is shown as the orange curve in Figure 3 . We create a histogram of the norm of its first 3 dimensions (Figure 1 (a)) and found that more than half of the generated data falls inside of the unit sphere. The generated data has one intrinsic dimension higher than its training data, despite its decoder variance approaching zero, which is equivalent to its reconstruction error approaching zero by Lemma 2.

In the sigmoid dataset, the featured model has 24 latent dimension, and is trained on a 7-dimensional manifold embedded in a 28-dimensional ambient space. We produced a scatter plot given 1000 generated data points $\tilde{x}_{r+1}$ from the decoder. The $x$-axis in the Figure 1(b) is $\langle a^*, \tilde{x}_{:r} \rangle$ and the $y$-axis is $\tilde{x}_{r+1}$. In contrast to the groundtruth data, whose scatter points roughly form a sigmoid function, the scatter points of the generate data resemble a gaussian distribution. This closely resembles the example provided in Theorem 6. Hence, despite its decoder variance and reconstruction error both approaching zero and loss consistently decreasing, the generated data do not recover the training data distribution and the data distribution recovered has higher intrinsic dimensions than the training data. We also investigated the effect of lower bounding the decoder variance as a possible way to improve the VAE performance (details are given in Appendix E). This enabled the VAE to recover the correct manifold dimension in the sigmoid example, but not the sphere example; methods of improvements to the VAE's manifold recovery is an important direction for future work.

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

## A  DERIVATIONS OF VAE LOSSES

We have (for some constants $C_1, C_2, C_3$):

$$\log p(x|z) = -\frac{1}{2\epsilon^2}\|x - f(z)\|^2 - d\log(\epsilon) + C_1$$

$$\log p(z) = -\|z\|^2/2 + C_2$$

$$\log q(z|x) = -\frac{1}{2}\langle z - g(x), D^{-1}(z - g(x))\rangle - \log\sqrt{\det D} + C_3$$

where the first line uses that $\log\sqrt{\det \epsilon^2 I} = \log\sqrt{\epsilon^{2d}} = d\log(\epsilon)$. The VAE objective is to maximize the expectation of $\log p(x|z) + \log p(z) - \log q(z|x)$ for $x$ from the data $p^*$ and $z \sim q(z|x)$. This means that explicitly the objective is to *maximize*

$$\mathbb{E}_{x\sim p^*}\mathbb{E}_{z\sim q(z|x)}\left[\log p(x|z) + \log p(z) - \log q(z|x)\right] - C$$

$$= \mathbb{E}_{x\sim p^*}\mathbb{E}_{z\sim q(z|x)}\left[-\frac{1}{2\epsilon^2}\|x - f(z)\|^2 - d\log(\epsilon) - \|z\|^2/2 + \frac{1}{2}\langle z - g(x), D^{-1}(z - g(x))\rangle + \log\sqrt{\det D}\right]$$

$$= \mathbb{E}_{x\sim p^*}\mathbb{E}_{z'\sim N(0,I_r)}\left[-\frac{1}{2\epsilon^2}\|x - f(g(x) + D^{1/2}z')\|^2 - d\log(\epsilon) - \|g(x) + D^{1/2}z'\|^2/2\right.$$
$$\left. + \frac{1}{2}\langle z', z'\rangle + \log\sqrt{\det D}\right]$$

which simplifies to (up to additive constant)

$$\mathbb{E}_{x\sim p^*}\mathbb{E}_{z'\sim N(0,I_r)}\left[-\frac{1}{2\epsilon^2}\|x - f(g(x) + D^{1/2}z')\|^2 - \|g(x)\|^2/2\right] - d\log(\epsilon) - \text{Tr}(D)/2 + \frac{1}{2}\sum_i \log D_{ii}.$$

and converting this to minimization form gives the VAE Loss (3).

**Linear VAE derivation.**  Plugging in the linear VAE parameters into the loss function, we get

$$L(\tilde{A}, \tilde{B}, \tilde{D}, \tilde{\epsilon}) := \mathbb{E}_{x\sim p^*}\mathbb{E}_{z'\sim N(0,I_{\tilde{r}})}\left[\frac{1}{2\tilde{\epsilon}^2}\|x - \tilde{A}(\tilde{B}x + \tilde{D}^{1/2}z')\|^2 + \|\tilde{B}x\|^2/2\right] \tag{7}$$

$$+ d\log(\tilde{\epsilon}) + \text{Tr}(\tilde{D})/2 - \frac{1}{2}\sum_i \log\tilde{D}_{ii} \tag{8}$$

We can write out the expectation as:

$$\mathbb{E}_{z\sim N(0,I)}\mathbb{E}_{z'\sim N(0,I_{\tilde{r}})}\left[\frac{1}{2\tilde{\epsilon}^2}\|Az - \tilde{A}(\tilde{B}Az + \tilde{D}^{1/2}z')\|^2 + \|\tilde{B}Az\|^2/2\right]$$

$$= \mathbb{E}_{z\sim N(0,I)}\mathbb{E}_{z'\sim N(0,I_{\tilde{r}})}\left[\frac{1}{2\tilde{\epsilon}^2}\|(A - \tilde{A}\tilde{B}A)z - \tilde{A}\tilde{D}^{1/2}z'\|^2 + \|\tilde{B}Az\|^2/2\right]$$

$$= \frac{1}{2\tilde{\epsilon}^2}\|A - \tilde{A}\tilde{B}A\|_F^2 + \frac{1}{2\tilde{\epsilon}^2}\|\tilde{A}\tilde{D}^{1/2}\|_F^2 + \frac{1}{2}\|\tilde{B}A\|_F^2$$

where we used that $z, z'$ are independent and the identity $\mathbb{E}_{z\sim N(0,I)}\|Mz\|^2 = \langle MM^T, I\rangle = \|M\|_F^2$. Next, we can observe that

$$\|\tilde{A}\tilde{D}^{1/2}\|_F^2 = \sum_i \tilde{D}_{ii}\|\tilde{A}_i\|^2$$

where $\tilde{A}_i$ is the $i$-th column of the matrix $\tilde{A}$. Therefore we recover (4).

# B   DEFERRED PROOFS FROM SECTION 4

## B.1   GENERAL FACTS

*Proof of Lemma 1.* For completeness, we include the proof of these claims; they are similar to the proofs of Theorems 4 and 5 in Dai & Wipf (2019).

First, consider the objective for fixed $f, g, D, \epsilon$ and omit the subscript $t$. We have

$$\mathbb{E}_{x \sim p^*} \mathbb{E}_{z' \sim N(0, I_r)} \left[ \frac{1}{2\epsilon^2} \|x - f(g(x) + D^{1/2} z')\|^2 + \|g(x)\|^2/2 \right] \geq 0$$

and

$$\mathrm{Tr}(D)/2 - \frac{1}{2} \sum_i \log D_{ii} = \frac{1}{2} \sum_i (D_{ii} - \log D_{ii}) \geq r/2$$

from the inequality $x - \log x \geq 1$ for $x \geq 0$. Since these terms are both bounded below, the only way the objective goes to negative infinity is if $d \log \epsilon \to -\infty$ which means $\epsilon \to 0$.

Now that we know $\epsilon_t \to 0$, we claim that $\lim_{t \to \infty} \mathbb{E}_{x \sim p^*} \mathbb{E}_{z' \sim N(0, I_r)} \|x - f_t(g_t(x) + D_t^{1/2} z')\|^2 = 0$. Suppose otherwise: then this for infinitely many $t$ this quantity is lower bounded by some constant $c > 0$, hence the objective for those $t$ is lower bounded by $c/\epsilon^2 + d \log(\epsilon) + r/2$ and this goes to $+\infty$ as $\epsilon \to 0$, instead of $-\infty$. □

*Proof of Lemma 2.* Taking the partial derivative of (3) with respect to $\epsilon$ and setting it to zero gives

$$0 = -\frac{1}{\epsilon^3} \mathbb{E}_{x \sim p^*} \mathbb{E}_{z' \sim N(0, I_r)} \|x - f(g(x) + D^{1/2} z')\|^2 + \frac{d}{\epsilon}$$

and solving for $\epsilon$ gives the result. □

## B.2   LINEAR VAE

*Proof of Lemma 3.* Taking the partial derivative with respect to $\tilde{D}_{ii}$ gives $0 = \|\tilde{A}_i\|^2/\tilde{\epsilon}^2 + 1 - 1/\tilde{D}_{ii}$ which means

$$\tilde{D}_{ii} = \frac{1}{\|\tilde{A}_i\|^2/\tilde{\epsilon}^2 + 1} = \frac{\tilde{\epsilon}^2}{\|\tilde{A}_i\|^2 + \tilde{\epsilon}^2}$$

hence

$$\tilde{D}_{ii} \|\tilde{A}_i\|^2/\tilde{\epsilon}^2 + \tilde{D}_{ii} - \log \tilde{D}_{ii} = 1 - \log \frac{\tilde{\epsilon}^2}{\|\tilde{A}_i\|^2 + \tilde{\epsilon}^2}.$$

It follows that the objective value at the optimal $D$ is

$$L_1(\tilde{A}, \tilde{B}, \tilde{\epsilon}) = \frac{1}{2\tilde{\epsilon}^2} \|A - \tilde{A}\tilde{B}A\|_F^2 + \frac{1}{2} \|\tilde{B}A\|_F^2 + d \log \tilde{\epsilon} + \frac{1}{2} \sum_i \left( 1 - \log \frac{\tilde{\epsilon}^2}{\|\tilde{A}_i\|^2 + \tilde{\epsilon}^2} \right)$$

$$= \frac{1}{2\tilde{\epsilon}^2} \|A - \tilde{A}\tilde{B}A\|_F^2 + \frac{1}{2} \|\tilde{B}A\|_F^2 + (d - r) \log \tilde{\varepsilon} + \frac{1}{2} \sum_i \left( 1 - \log \frac{1}{\|\tilde{A}_i\|^2 + \tilde{\epsilon}^2} \right).$$

□

*Proof of Theorem 3.* First we prove the sufficiency direction, i.e. that if $A = \tilde{A}\tilde{B}A$ and there exists $i$ such that $\tilde{A}_i = 0$ then we show how to drive the loss to $-\infty$. By Lemma 3, if we make the optimal choice of $D$ (which clearly satisfies the conditions on $D$ described in the Lemma) the objective simplifies to

$$L_1(\tilde{A}, \tilde{B}, \tilde{\epsilon}) = \frac{1}{2\tilde{\epsilon}^2} \|A - \tilde{A}\tilde{B}A\|_F^2 + \frac{1}{2} \|\tilde{B}A\|_F^2 + (d - r) \log \tilde{\varepsilon} + \frac{1}{2} \sum_i \left( 1 + \log \left( \|\tilde{A}_i\|^2 + \tilde{\epsilon}^2 \right) \right)$$

$$= \frac{1}{2} \|\tilde{B}A\|_F^2 + (d - r) \log \tilde{\varepsilon} + \frac{1}{2} \sum_i \left( 1 + \log \left( \|\tilde{A}_i\|^2 + \tilde{\epsilon}^2 \right) \right)$$

where in the second line we used the assumption $A = \tilde{A}\tilde{B}A$. Note that for each zero column $\tilde{A}_i = 0$ we have $(1/2)\log(\|\tilde{A}_i\|^2 + \tilde{\epsilon}^2) = \log\tilde{\epsilon}$ so the objective will go to $-\infty$ provided $(d - r + \#\{i : \tilde{A}_i = 0\})\log\tilde{\epsilon} \to -\infty$. Since $\tilde{\epsilon} \to 0$ this is equivalent to asking $d - r + \#\{i : \tilde{A}_i = 0\} > 0$, which is exactly the assumption of the Theorem.

Next we prove the converse direction, i.e. the necessary conditions. Note: we split the first item in the lemma into two conclusions in the proof below (so there are four conclusions instead of three). The first conclusion follows from the first conclusion of Lemma 1. The second conclusion of Lemma 1 tells us that

$$0 = \lim_{t\to\infty} \mathbb{E}_{z\sim N(0,I)}\mathbb{E}_{z'\sim N(0,I_{\tilde{r}})}\|Az - \tilde{A}_t(\tilde{B}_tAz + \tilde{D}_t^{1/2}z')\|^2 = \lim_{t\to\infty}\|A - \tilde{A}_t\tilde{B}_tA\|_F^2 + \|\tilde{A}_t\tilde{D}_t^{1/2}\|_F^2$$

which gives us the second and third conclusions above. For the fourth conclusion, since $L_1(\tilde{A}_t, \tilde{B}_t, \tilde{D}_t) \leq L(\tilde{A}_t, \tilde{B}_t, \tilde{D}_t, \tilde{\epsilon}_t)$ we know that $\lim_{t\to\infty} L_1(\tilde{A}_t, \tilde{B}_t, \tilde{D}_t) = -\infty$ and recalling

$$L_1(\tilde{A}, \tilde{B}, \tilde{\epsilon}) = \frac{1}{2\tilde{\epsilon}^2}\|A - \tilde{A}\tilde{B}A\|_F^2 + \frac{1}{2}\|\tilde{B}A\|_F^2 + (d-r)\log\tilde{\epsilon} + \frac{1}{2}\sum_i\left(1 + \log\left(\|\tilde{A}_i\|^2 + \tilde{\epsilon}^2\right)\right)$$

we see that, because the first two terms are nonnegative, this is possible only if the sum of the last two terms goes to $-\infty$. Based on similar reasoning to the sufficiency case, this is only possible if strictly more than $r - d$ of the columns of $(\tilde{A}_t)$ become arbitrarily close to zero; precisely, if there exists $\delta$ such that at most $r - d$ of the columns of $\tilde{A}_t$ have norm less than $\delta$, then

$$(d-r)\log\tilde{\epsilon} + \frac{1}{2}\sum_i\left(1 + \log\left(\|\tilde{A}_i\|^2 + \tilde{\epsilon}^2\right)\right)$$

$$\geq \frac{1}{2}\sum_{i:\|\tilde{A}_i\|\geq\delta}\left(1 + \log\left(\|\tilde{A}_i\|^2 + \tilde{\epsilon}^2\right)\right)$$

$$\geq \frac{1}{2}\sum_{i:\|\tilde{A}_i\|\geq\delta}\left(1 + \log\left(\delta^2 + \tilde{\epsilon}^2\right)\right)$$

which does not go to $-\infty$ as $\tilde{\epsilon} \to 0$ (and the other terms of $L_1$ are nonnegative). $\qquad\square$

### B.3   Nonlinear VAE

We give the full details of the construction of the bad nonlinear VAE optimum and prove that it is an asymptotic global minimum. (Note: in the notation of Section 6 we are considering $a^*$ with 0/1 entries, but the proof generalizes straightforwardly for arbitrary $a^*$ with the correct support.)

**Setup:**   Suppose $s \geq 1$ is arbitrary and the ground truth $x \in \mathbb{R}^d$ with $d > r^* + s$ is generated in the following way: $(x_1, \ldots, x_{r^*}) \sim N(0, I_{r^*})$, $x_{r^*+1} = \sigma(x_1 + \cdots + x_{r^*})$ for an arbitrary nonlinearity $\sigma$, and $x_{r^*+2} = \cdots = x_d = 0$. Furthermore, suppose the architecture for the decoder with latent dimension $r > r^* + 1$ is

$$f_{\tilde{A}_1, \tilde{A}_2}(z) := \tilde{A}_1 z + \sigma\left(\tilde{A}_2 z\right)$$

where $\sigma(\cdot)$ is applied as an entrywise nonlinearity, and the encoder is linear, $g(x) := \tilde{B}x$.

Observe that the ground truth decoder can be expressed by taking $\tilde{A}_2$ to have a single nonzero row in position $r + 1$ with entries $(1, \ldots, 1, 0, \ldots, 0)$,

$$\tilde{A}_1 = \begin{bmatrix} I_{r^*} & 0 \\ 0 & 0 \end{bmatrix}, \quad \tilde{B} = \begin{bmatrix} I_{r^*} & 0 \\ 0 & 0 \end{bmatrix}.$$

where $\tilde{B}$ is a ground truth encoder which achieves perfect reconstruction.

On the other hand, the following VAE different from the ground truth achieves perfect reconstruction:

$$\tilde{A}_1 = \begin{bmatrix} I_{r^*+s} & 0 \\ 0 & 0 \end{bmatrix}, \quad \tilde{A}_2 = 0, \quad \tilde{B} = \begin{bmatrix} I_{r^*+1} & 0 \\ 0 & 0 \end{bmatrix} \tag{9}$$

The output of this decoder is a Gaussian $N\left(0, \begin{bmatrix} I_{r^*+s} & 0 \\ 0 & 0 \end{bmatrix}\right)$, which means it is strictly higher-dimensional than the ground truth dimension $r^*$. (This also means that if we drew the corresponding plot of to Figure 1 (b) for this model, we would get something that looks just like the experimentally obtained result.) We prove in the Appendix that it this is an asymptotic global optima:

**Theorem 6.** *Let $s \geq 1$ be arbitrary and the ground truth and VAE architecture is as defined as above. For any sequence $\tilde{\epsilon}_t \to 0$, there exist diagonal matrices $\tilde{D}_t$ such that:*

1. *the VAE loss $L(\tilde{A}_1, \tilde{A}_2, \tilde{B}, \tilde{D}_t, \tilde{\epsilon}_t) \to -\infty$ where $\tilde{A}_1, \tilde{A}_2, \tilde{B}$ are defined by* (9)

2. *The number of coordinates of $\tilde{D}_t$ which go to zero equals $r^* + s$.*

*Proof.* We show how to pick $\tilde{D}_t$ as a function of $\tilde{\epsilon}_t$ and that if $\tilde{\epsilon}_t \to 0$, the loss goes to $-\infty$. From now on we drop the subscripts.

With these parameters, the VAE loss is

$$\mathbb{E}_{x \sim p^*} \mathbb{E}_{z' \sim N(0, I_r)} \left[ \frac{1}{2\tilde{\epsilon}^2} \|x - f(g(x) + \tilde{D}^{1/2} z')\|^2 + \|g(x)\|^2/2 \right] + d \log(\tilde{\epsilon}) + \text{Tr}(\tilde{D})/2 - \frac{1}{2} \sum_i \log \tilde{D}_{ii}$$

$$= (1/2\tilde{\epsilon}^2) \sum_{i=1}^{r^*+1} \tilde{D}_{ii} + E_{x \sim p^*} \left[ \|x_{1:r^*+1}\|^2/2 \right] + d \log(\tilde{\epsilon}) + \text{Tr}(\tilde{D})/2 - \frac{1}{2} \sum_i \log \tilde{D}_{ii}.$$

Taking the partial derivative with respect to $\tilde{D}_{ii}$ for $i \leq r^* + s$ and optimizing gives $0 = (1/\tilde{\epsilon}^2) + 1 - 1/\tilde{D}_{ii}$ i.e.

$$\tilde{D}_{ii} = \frac{1}{1 + 1/\tilde{\epsilon}^2} = \frac{\tilde{\epsilon}^2}{\tilde{\epsilon}^2 + 1}$$

and plugging this into the objective gives

$$(1/2\tilde{\epsilon}^2) \sum_{i=1}^{r^*+1} \tilde{D}_{ii} + E_{x \sim p^*} \left[ \|x_{1:r^*+1}\|^2/2 \right] + d \log(\tilde{\epsilon}) + \text{Tr}(\tilde{D})/2 - \frac{1}{2} \sum_i \log \tilde{D}_{ii}$$

$$= (1/2) \sum_{i=1}^{r^*+1} \frac{1}{\tilde{\epsilon}^2 + 1} + E_{x \sim p^*} \left[ \|x_{1:r^*+1}\|^2/2 \right] + (d - r^* - s) \log(\tilde{\epsilon})$$

$$+ \text{Tr}(\tilde{D})/2 + \frac{r^* + 1}{2} \log(1 + \epsilon^2) + \frac{1}{2} \sum_{i=r^*+2}^{r} \log \tilde{D}_{ii}.$$

Setting the remaining $\tilde{D}_{ii}$ to 1, we see that using $d > r^* + s$ that the loss goes to $-\infty$ provided $\tilde{\epsilon} \to 0$, proving the result. $\qquad \square$

## C  DEFERRED PROOFS FROM SECTION 5

First, we formalize the rotation invariance of the objective.

**Lemma 6** (Rotational Invariance of Gradient Descent on Linear VAE). *Let $L_A(\tilde{A}, \tilde{B}, \tilde{D}, \tilde{\epsilon})$ denote the VAE population loss objective* (4). *Then for an arbitrary orthogonal matrix $U$, we have*

$$L_A(\tilde{A}, \tilde{B}, \tilde{D}, \tilde{\epsilon}) = L_{UA}(U\tilde{A}, \tilde{B}U^T, \tilde{D}, \tilde{\epsilon}).$$

*Furthermore,*

$$U \nabla_{\tilde{A}} L_A(\tilde{A}, \tilde{B}, \tilde{D}, \tilde{\epsilon}) = \nabla_{U\tilde{A}} L_{UA}(U\tilde{A}, \tilde{B}U^T, \tilde{D}, \tilde{\epsilon})$$

*and*

$$(\nabla_{\tilde{B}} L_A(\tilde{A}, \tilde{B}, \tilde{D}, \tilde{\epsilon})) U^T = \nabla_{U\tilde{B}} L_{UA}(U\tilde{A}, \tilde{B}U^T, \tilde{D}, \tilde{\epsilon}).$$

*As a consequence, if for any $\eta \geq 0$ we define $(\tilde{A}_1, \tilde{B}_1, \tilde{D}_1, \tilde{\epsilon}_1) = (\tilde{A}, \tilde{B}, \tilde{D}, \tilde{\epsilon}) - \eta \nabla L_A(\tilde{A}, \tilde{B}, \tilde{D}, \tilde{\epsilon})$ then*

$$(U\tilde{A}_1, \tilde{B}_1 U^T, \tilde{D}_1, \tilde{\epsilon}_1) = (U\tilde{A}, \tilde{B}U^T, \tilde{D}, \tilde{\epsilon}) - \eta \nabla_{(U\tilde{A}, \tilde{B}U^T, \tilde{D}, \tilde{\epsilon})} L_{UA}(U\tilde{A}, \tilde{B}U^T, \tilde{D}, \tilde{\epsilon}),$$

*i.e. gradient descent preserves rotations by $U$. The same result holds for the gradient flow (i.e. continuous time gradient descent), or replacing everywhere the loss $L$ by the simplified loss $L_1$.*

*Proof of Lemma 6.* We give the proof for $L$ as stated, but it is exactly the same for the simplified loss $L_1$.

From the objective function (4) and $U^T = U^{-1}$ observe that

$$L_{UA}(U\tilde{A}, \tilde{B}U^T, \tilde{D}, \tilde{\epsilon})$$
$$= \frac{1}{2\tilde{\epsilon}^2}\|UA - U\tilde{A}\tilde{B}U^{-1}UA\|_F^2 + \frac{1}{2}\|\tilde{B}U^{-1}UA\|_F^2 + d\log\tilde{\epsilon} + \frac{1}{2}\sum_i \left(\tilde{D}_{ii}\|U\tilde{A}_i\|^2/\tilde{\epsilon}^2 + \tilde{D}_{ii} - \log\tilde{D}_{ii}\right)$$
$$= \frac{1}{2\tilde{\epsilon}^2}\|A - \tilde{A}\tilde{B}A\|_F^2 + \frac{1}{2}\|\tilde{B}A\|_F^2 + d\log\tilde{\epsilon} + \frac{1}{2}\sum_i \left(\tilde{D}_{ii}\|\tilde{A}_i\|^2/\tilde{\epsilon}^2 + \tilde{D}_{ii} - \log\tilde{D}_{ii}\right)$$
$$= L_A(\tilde{A}, \tilde{B}, \tilde{D}, \tilde{\epsilon}).$$

Then from the above and the multivariate chain rule have

$$\nabla_{\tilde{A}}L_A(\tilde{A}, \tilde{B}, \tilde{D}, \tilde{\epsilon}) = \nabla_{\tilde{A}}L_{UA}(U\tilde{A}, \tilde{B}U^T, \tilde{D}, \tilde{\epsilon}) = U^T\left(\nabla_{U\tilde{A}}L_{UA}(U\tilde{A}, \tilde{B}U^{-1}, \tilde{D}, \tilde{\epsilon})\right)$$

so multiplying both sides on the left by $U$ and using $U^T = U^{-1}$ gives the second claim, and similarly

$$\nabla_{\tilde{B}}L_A(\tilde{A}, \tilde{B}, \tilde{D}, \tilde{\epsilon}) = \nabla_{\tilde{B}}L_{UA}(U\tilde{A}, \tilde{B}U^T, \tilde{D}, \tilde{\epsilon}) = (\nabla_{\tilde{B}U^T}L_{UA}(U\tilde{A}, \tilde{B}U^T, \tilde{D}, \tilde{\epsilon}))U$$

gives the third claim. Then the gradient descent claim follows immediately. $\square$

*Proof of Lemma 4.* First we prove the conclusion for the original loss $L$. Since $(\tilde{A}\tilde{B}A)_{i\ell} = \sum_{j,k}\tilde{A}_{ij}\tilde{B}_{jk}A_{k\ell}$ we have that

$$\frac{\partial\|A - \tilde{A}\tilde{B}A\|_F^2}{\partial\tilde{A}_{ij}} = \frac{\partial}{\partial\tilde{A}_{ij}}\sum_\ell\left(A_{i\ell} - \sum_{j',k}\tilde{A}_{ij'}\tilde{B}_{j'k}A_{k\ell}\right)^2 = \sum_\ell 2\left(A_{i\ell} - \sum_{j',k}\tilde{A}_{ij'}\tilde{B}_{j'k}A_{k\ell}\right)\left(-\sum_k\tilde{B}_{jk}A_{k\ell}\right)$$

and if we know the corresponding row $i$ in $A$ is zero then this simplifies to

$$\frac{\partial\|A - \tilde{A}\tilde{B}A\|_F^2}{\partial\tilde{A}_{ij}} = \sum_\ell 2\left(\sum_{j',k}\tilde{A}_{ij'}\tilde{B}_{j'k}A_{k\ell}\right)\left(\sum_k\tilde{B}_{jk}A_{k\ell}\right)$$

which means that

$$\sum_j\tilde{A}_{ij}\frac{\partial\|A - \tilde{A}\tilde{B}A\|_F^2}{\partial\tilde{A}_{ij}} = \sum_\ell 2\left(\sum_{j,k}\tilde{A}_{ij}\tilde{B}_{jk}A_{k\ell}\right)^2 = 2\|(\tilde{A}\tilde{B}A)^{(i)}\|^2$$

where the notation $A^{(i)}$ denotes row $i$ of matrix $A$. Thus, for this term the gradient with respect to row $\tilde{A}^{(i)}$ has nonnegative dot product with row $\tilde{A}^{(i)}$.

Also,

$$\frac{\partial}{\partial\tilde{A}_{ij}}(1/2)\sum_i\tilde{D}_{jj}\|\tilde{A}_j\|^2/\tilde{\epsilon}^2 = \tilde{D}_{jj}\tilde{A}_{ij}/\tilde{\epsilon}^2$$

and so

$$\sum_j\tilde{A}_{ij}\frac{\partial}{\partial\tilde{A}_{ij}}(1/2)\sum_j\tilde{D}_j\|\tilde{A}_j\|^2/\tilde{\epsilon}^2 = \sum_j\tilde{D}_{jj}\tilde{A}_{ij}^2/\tilde{\epsilon}^2$$

which gives the first result.

For the second result with the simplified loss $L_1$, observe that

$$\frac{\partial}{\partial\tilde{A}_{ij}}\sum_k\log(\|\tilde{A}_k\|^2 + \tilde{\epsilon}^2) = \frac{2\tilde{A}_{ij}}{\|\tilde{A}_j\|^2 + \tilde{\epsilon}^2}$$

so

$$\sum_j\tilde{A}_{ij}\frac{\partial}{\partial\tilde{A}_{ij}}\sum_k\log(\|\tilde{A}_k\|^2 + \tilde{\epsilon}^2) = \sum_j\frac{2\tilde{A}_{ij}^2}{\|\tilde{A}_j\|^2 + \tilde{\epsilon}^2}$$

and the other terms in the loss behave the same in the case of $L$. Including the factor of $1/2$ from the loss function gives the result. $\square$

*Proof of Lemma 5.* From Lemma 4 we have that for any such row $i$,

$$\frac{d}{dt}\|\tilde{A}^{(i)}(t)\|^2 = 2\langle \tilde{A}^{(i)}(t), \frac{d}{dt}\tilde{A}^{(i)}(t)\rangle$$

$$= 2\langle \tilde{A}^{(i)}(t), -\nabla_{\tilde{A}(t)^{(i)}}L_1(\Theta_t)\rangle \leq -\sum_{j=1}^{r}\frac{(\tilde{A}(t))_{ij}^2}{\|(\tilde{A}(t))_j\|^2 + \tilde{\epsilon}_t^2} \leq -(1/K)\|\tilde{A}^{(i)}(t)\|^2$$

which by Gronwall's inequality implies $\|\tilde{A}^{(i)}(t)\|^2 \leq e^{-t/K}\|\tilde{A}^{(i)}(0)\|^2$ as desired. $\qquad\square$

*Proof of Theorem 5.* Before proceeding, we observe that the first inequality in (6) is a consequence of the general min-max characterization of singular values, see e.g. Horn & Johnson (2012). We now prove the rest of the statement.

As explained at the beginning of the section, we start by taking the Singular Value Decomposition $A = USV^T$ where $S$ is the diagonal matrix of singular values and $U, V$ are orthogonal. We assume the diagonal matrix $S$ is sorted so that its top-left entry is the largest singular value and its bottom-right is the smallest. Note that this means the first $\dim(W)$ rows of $U$ are an orthonormal basis for $W$. Note that for any time $t$, $\|P_{W^\perp}\tilde{A}^T(t)\|_F^2 = \sum_{i=\dim(W)+1}^{d}\|(U\tilde{A}(t)^T)_i\|^2$ because the rows $(U_{\dim(W)+1}, \ldots, U_d)$ are an orthonormal basis for $W^\perp$. Therefore we have that

$$\|P_{W^\perp}\tilde{A}^T(t)\|_F^2 = \sum_{i=\dim(W)+1}^{d}\|(U\tilde{A}(t)^T)_i\|^2$$

$$\leq e^{-t/K}\sum_{i=\dim(W)+1}^{d}\|(U\tilde{A}(0)^T)_i\|^2 = e^{-t/K}\|P_{W^\perp}\tilde{A}^T(0)\|_F^2,$$

proving the result, provided we justify the middle inequality. Define $A^* := U^T A = SV^T$, which has a zero row for every zero singular value of $A$, and apply Lemma 5 (using that the definition of $K$ is invariant to left-multiplication of $\tilde{A}$ by an orthogonal matrix) and Lemma 6 to conclude that the rows of $U^T\tilde{A}(t)$, i.e. the columns of $U\tilde{A}(t)^T$, corresponding to zero rows of $A^*$ shrink by a factor of $e^{-t/K}$. This directly gives the desired inequality, completing the proof. $\qquad\square$

# D    DEFERRED FIGURES AND PLOTS FROM SECTION 6

**Eigenvalues of Linear Data.**    As we've discussed, in our linear setting the VAE does not recover the ground truth data density. Since our generative process for ground-truth data is $x = Az$ for a matrix $A$ and $z$ normally distributed, we can characterize the density function by the eigenvalues of the true or estimated covariance matrix. We give figures for the normalized error of these eigenvalues between the learned generator and the ground truth in Table 1. A concrete example of eigenvalue mismatch for a problem with 6 nonzero dimensions is a ground-truth set of covariance eigenvalues

$$\lambda = [0.001 \quad 0.156 \quad 1.54 \quad 5.06 \quad 9.55 \quad 16.4]$$

while the trained linear VAE distribution has covariance eigenvalues

$$\hat{\lambda} = [0.035 \quad 0.166 \quad 1.49 \quad 4.24 \quad 5.97 \quad 7.85].$$

Here, the VAE was easily able to learn the support of the data but clearly is very off when it comes to the structure of the covariances.

# E    EXPERIMENTS WITH DECODER VARIANCE CLIPPING

As was suggested by an anonymous ICLR 2022 reviewer, one potential way to evade the results in our paper is to restrict the decoder variance from converging to 0. In this section, we examine (empirically) the impact of clipping the decoder variance during training. We caveat though, that our paper does not analyze the landscape of the resulting constrained optimization problem, so our results don't imply anything about this regime.

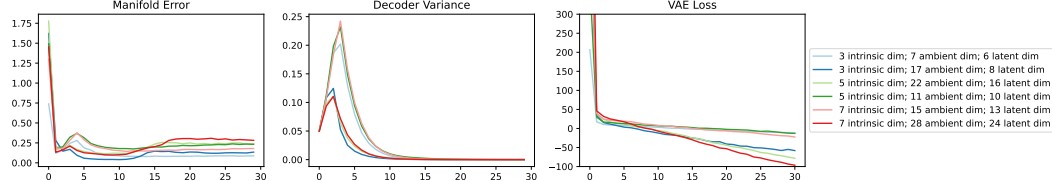

Figure 2: VAE training on 6 datasets with different choices of dimensions for sigmoidal dataset (see Setup in Section 6). The $x$-axis represents every 5000 gradient updates during training. The left-most figure is the manifold error (see Setup in Section 6), The middle and right figure confirms that the decoder variance approaches zero and the VAE loss is steadily decreasing during the finite training time.

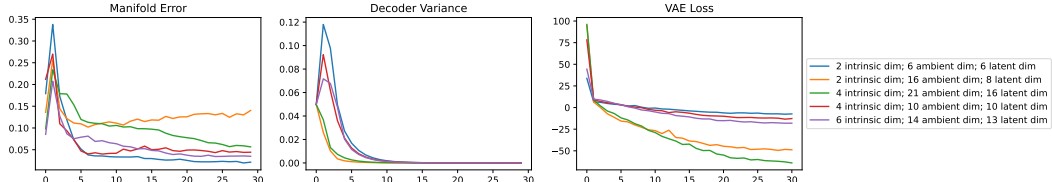

Figure 3: VAE training on 5 datasets generated by appending zeros to uniformly random samples from a unit sphere to embed in a higher dimensional ambient space. The $x$-axis represents each iteration of every 5000 gradient updates. The left-most figure is the manifold error ( see Setup in Section 6), The middle and right figure confirms that the decoder variance approaches zero and the VAE loss is steadily decreasing during the finite training time.

| Intrinsic Dimensions | 2 | 2 | 4 | 4 | 6 |
|---|---|---|---|---|---|
| Ambient Dimensions | 6 | 16 | 10 | 21 | 14 |
| VAE Latent Dimensions | 6 | 8 | 10 | 16 | 13 |
| Mean Manifold Error | 0.02 | 0.14 | 0.04 | 0.06 | 0.03 |
| Mean #0's in Encoder Variance | 3 | 5 | 5 | 6 | 7 |

Table 3: Optima found by training a VAE on data generated by padding uniformly random samples from a unit $r$-sphere with zeroes, so that the sphere is embedded in a higher ambient dimension. We evaluated the manifold error as described in the setup. The VAE training on this dataset has consistently yielded encoder variances with number of 0 entries greater than the number of intrinsic dimension.

We conduct the same nonlinear experiments described in Section 6 where we fit VAEs to data generated from spheres and linear sigmoid functions. The only change is to clip the decoder variance when it goes below a certain threshold. In the figures below, the chosen threshold is $e^{-4} \approx 0.018$, though we tried also $e^{-2}$ and $e^{-3}$, with similar outcomes. We initialize the decoder variance with $e^{-3}$ for this set of experiments, so the optimization still can decrease it.

With this change, the optimization process on the sigmoid dataset does yield encoder variances with their number of zeros reflective of their intrinsic dimensions as in Table 4. For the sphere experiment, this still does not happen, as in Table 5. In fact, the model consistently recovers one more dimension than the true intrinsic dimension of the manifold and the smaller encoder variances can be as large as 0.1. We also provide a figure (Figure 4) in the same style as Figure 1. We see that training with a clipped decoder variance allows the model to better capture the general shape of the sigmoid function, though the variance of the generated points is high for both of the sphere and sigmoid datasets. Other training details, such as the general trend of manifold error, encoder variance and VAE loss, can be referred to in Figure 5 and 6.

Overall, the benefit of clipping the decoder variance during training is inconclusive as we see inconsistent results in the sphere and sigmoid datasets. Designing more algorithms to improve the ability of VAE's to recover data supported on a low dimensional manifold is an important direction for future work—both empirical and theoretical.

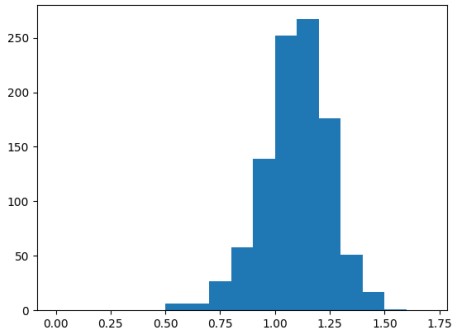 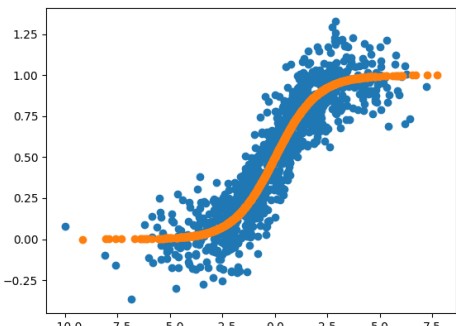

Figure 4: A demonstration of how the data points generated by the model trained with clipped decoder variance is distributed. *Left figure:* A histogram of the norms of samples generated from the VAE restricted to the dimensions which are not zero, which shows many of the points have norm less than 1. (The ground-truth distribution would output only samples of norm 1.) The particular example here is Column 2 in Table 5. The data points that do not fall on the sphere tend to lie on both sides of it whereas the those generated without decoder variance clipping tend to lie inside the sphere as in Figure 1. *Right figure:* Two-dimensional linear projection of data output by VAE generator trained on our sigmoid dataset. The $x$-axis denotes $\langle a^*, \tilde{x}_{:r} \rangle$ and the $y$-axis is $\tilde{x}_{r+1}$, the blue points are from the trained VAE and the orange points are from the ground truth. The generated data points roughly capture the shape of the sigmoid function.

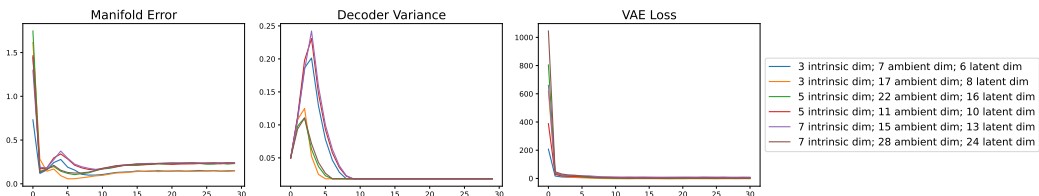

Figure 5: VAE training on 6 datasets with different choices of dimensions for sigmoidal dataset (see Setup in Section 6). The $x$-axis represents every 5000 gradient updates during training. The left-most figure is the manifold error (see Setup in Section 6), The middle and right figure shows that as the decoder variance is bounded below, the VAE loss stops decreasing further.

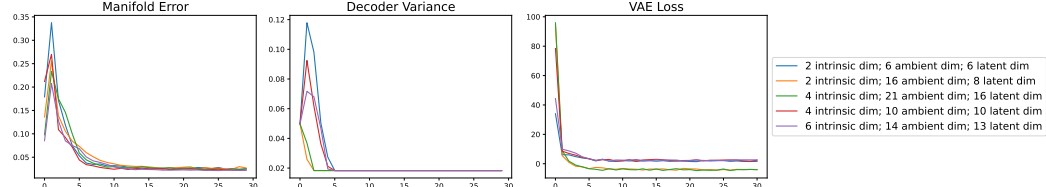

Figure 6: VAE training on 5 datasets generated by appending zeros to uniformly random samples from a unit sphere to embed in a higher dimensional ambient space. The $x$-axis represents each iteration of every 5000 gradient updates. The left-most figure is the manifold error ( see Setup in Section 6), The middle and right figure shows that as the decoder variance is bounded below, the VAE loss stops decreasing further.

| Intrinsic Dimensions | 3 | 3 | 5 | 5 | 7 | 7 |
|---|---|---|---|---|---|---|
| Ambient Dimensions | 7 | 17 | 11 | 22 | 15 | 28 |
| VAE Latent Dimensions | 6 | 8 | 10 | 16 | 13 | 24 |
| Mean Manifold Error | 0.15 | 0.15 | 0.23 | 0.23 | 0.24 | 0.24 |
| Mean #0's in Encoder Variance | 3 | 3 | 5 | 5 | 7 | 7 |

Table 4: Optima found by training a VAE on the sigmoid dataset. The VAE training yields encoder variances with number of 0 entries equal to the intrinsic dimension.

| Intrinsic Dimensions | 2 | 2 | 4 | 4 | 6 |
|---|---|---|---|---|---|
| Ambient Dimensions | 6 | 16 | 10 | 21 | 14 |
| VAE Latent Dimensions | 6 | 8 | 10 | 16 | 13 |
| Mean Manifold Error | 0.03 | 0.03 | 0.03 | 0.02 | 0.02 |
| Mean #0.1's in Encoder Variance | 3 | 3 | 5 | 5 | 7 |

Table 5: Optima found by training a VAE on data generated by padding uniformly random samples from a unit $r$-sphere with zeroes, so that the sphere is embedded in a higher ambient dimension. We evaluated the manifold error as described in the setup. The VAE training on this dataset has consistently yielded encoder variances with number of 0.1 entries greater than the number of intrinsic dimension.

