# OpenReview forum: "Variational autoencoders in the presence of low-dimensional data: landscape and implicit bias"
_ICLR.cc/2022/Conference — ICLR 2022 Poster_

### Official Review · Reviewer_U2NF · 2021-11-02

**Correctness:** 3
**Technical Novelty And Significance:** 3
**Empirical Novelty And Significance:** 3
**Recommendation:** 8
**Confidence:** 4

**Main Review:**

### Original Review

The results and informal discussions in Dai and Wipf (2019) can be misinterpreted in a few ways, and clarifications are valuable to the general ML audience.

That being said, most results here are arguably unsurprising, and largely stem from an unfortunate definition of "asymptotic optimality": it includes any (trajectory of) parameters which have loss tending to infinity, without accounting for the rate of divergence.  It is clear from e.g. Dai and Wipf (2019, p. 7) that any (suitable) VAE models parameterizing a $\hat{r}<d$ manifold will have its loss tend to infinity, as its leading term is $-\frac{d-\hat{r}}{2}\log \gamma$, where the decoder variance $\gamma$ is assumed to converges to zero.  It is also clear from the same source that if we lower bound $\gamma$ by a *small positive constant* $\gamma_0$, then (among the parameter sets considered in Dai and Wipf (2019) and this work) the global optima should have as small a dimension as possible, because the leading term of the loss will be $-\frac{d-\hat{r}}{2}\log \gamma_0$.  Dai and Wipf (2019) stated that their practical recommendations start from this form of the leading term of loss (last sentence of Section 3).  The result is relevant because in practice the optimal decoder variance will be bounded by the reconstruction error of the AE (Dai et al., 2020), although the theoretical picture is far from clear.

There are nuances around the informal discussions in Dai and Wipf (2019): one of the most important limitations is their restriction to a particular set of parameters (decoder variance is either 1 or near zero along each axis).  They also ignore errors from estimation, approximation and optimization.  However, the fact that there exists different parameter (trajectories) leading to model distributions with
support having different dimensions does not appear to be one of them.

The interesting part is the analysis of the implicit regularization effect from optimization, and the nonlinear simulation here reveals one surprising limitation of previous work that may have real consequences.  It would greatly enhance the paper if the authors experiment with simple fixes for this issue: e.g. clipping the decoder variance from below.  Regardless of whether the fixes work, the results will provide more insight as well as possible guidance for practitioners.

Minor comments:

1. In Theorem 4, $\tilde z$ is not defined; immediately below Theorem 5, it will be helpful to clarify the rotation is applied to both the model parameters and ground truth.

2. You may want to cite the following two papers, which improve the understanding of ELBO landscape over Dai and Wipf (2019):
    1. Kumar and Poole (2020), On Implicit Regularization in β-VAEs, ICML 2020.
    2. Tang and Yang (2021), On Empirical Bayes Variational Autoencoder: An Excess Risk Bound, COLT 2021.

### Post-rebuttal Update

My original concern is that the main results here appear irrelevant in light of the informal discussions in Dai and Wipf.  While I'm not sure if these discussions are not obvious, they do rely on strong assumptions and do not account for the optimization process.  More importantly, the findings in the new experiments provide contrary evidence to the discussions in Dai-Wipf.  This is interesting and demonstrates the relevance of the results in this work.  For this reason, I change my score to acceptance.

**Summary Of The Paper:**

This work revisits the population loss analysis for VAE ELBO (Dai and Wipf, 2019) and note that undesirable "asymptotic global optimas" exist, where the support of the model distribution $p_{model}(dx)$ has higher dimensions than the true data manifold.  Additionally, it shows that, for linear VAE, such optimas are excluded due to the implicit bias of gradient descent, but empirically nonlinear VAEs often stuck in such optimas.

**Summary Of The Review:**

(edited)

Strengths

+ The cautions against misinterpretation will be useful to the broad ML community.
+ Findings about the implicit bias are interesting.

Limitations

+ Further efforts are needed to understand the discrepancy between the results here, which do not provide rate of divergence, and the informal calculations in Dai and Wipf, which implicitly places assumptions that do not appear consistently supported by experiments.

---

> ### Author Response · Authors · 2021-11-13
> **Thank you for the detailed feedback!**
>
> Thank you for the detailed feedback! We hope this discussion can clear up your questions as well as improve the final version of the paper.
>
> **Regarding asymptotic optimality**: we are in agreement that part of our results have an asymptotic optimality flavor (Theorems 3 and 4, which characterize the set of optima in the linear and nonlinear case). Theorem 6 does however analyze the trajectory of the gradient dynamics, as you yourself note. We added some caveats regarding the nature of asymptotic results in Section 3.
>
> **Regarding “avoiding” the asymptotic regime**: If we understand your comment correctly, you are suggesting that the sequences of points whose loss decays to $-\infty$ at the fastest rate should recover the correct support. Also, you are suggesting restricting the decoder variance (in our notation) $\epsilon$ to be lower-bounded by some constant $\epsilon_0$ in the hope of making the only global minima of the resulting objective the intended ones.
>
> We would first caveat that nearly all of the above points are not obvious, even in the linear case, where our proofs are fully formal, unlike the heuristic calculation in Dai-Wipf. In the non-linear case, like you yourself note, the heuristic in Dai-Wipf assumes, among other things, that the entries in the covariance of the encoder are sharply delineated as tending to 0 or 1— and there is no a-priori reason to think this will happen.
>
> However, even assuming all this can be done, the landscape corresponding to the truncated variance loss will almost certainly have many different local minima and we would still (crucially!) need to analyze the behavior of the training dynamics to understand which ones gradient descent would converge to.
>
> We are currently running the experiments you suggested of training VAEs with clipped decoder variance and appreciate your suggestion greatly! We will include the results in the updated submission when we have finished.
>
> **Nuances in Dai-Wipf**: We agree, there are several big assumptions in Dai-Wipf. The most major one that we remove in our paper is the assumption that the encoder covariance asymptotically has either close to 0 or close to 1 eigenvalues.  Another key point, as the reviewer pointed out, is the role of the training dynamics which we explicitly address in this paper. When there are multiple global or local optima, as in the setting of this paper (and in all of its natural variants, e.g. with clipped decoder variance), analyzing the behavior of the training dynamics is the only reliable/canonical way to make a statement about which optima will be chosen in actual training.
>
> Thank you for your minor comments, we have looked into them and made the suggested changes.

---

> ### Author Response · Authors · 2021-11-22
> **Results of decoder variance clipping**
>
> Thanks again for all your suggestions. We have completed running our experiments in which we clip the decoder variance and included them in the paper in Appendix E.
>
> In brief, the results were inconclusive. On the sigmoid dataset, the number of (near)-zeros of the encoder variances correctly matched the intrinsic dimension. Moreover, as shown in Figure 4, the overall shape of the manifold is better captured. On the sphere dataset, this was unfortunately not the case.
>
> Thank you for the suggestion, and we will continue to look for empirical and theoretical modifications that will satisfyingly solve the problems we discuss in the paper.

---

### Official Review · Reviewer_MVZZ · 2021-11-02

**Correctness:** 3
**Technical Novelty And Significance:** 4
**Empirical Novelty And Significance:** 2
**Recommendation:** 6
**Confidence:** 3

**Main Review:**

The strengths of this paper are:
- Combination of loss landscape analysis with gradient flow analysis sounds sensible and validates the claim by (Dai & Wipf; 2019) for the linear case.
- The sigmoid dataset example is simple enough to see VAE may overestimate the intrinsic dimension.

The weaknesses are:
1) the organization of the paper could be improved, and
2) the paper lacks any tips for practitioners.

1: Section 3 explains the general ideas while Sections 4 and 5 give rigorous discussions about the loss and optimization dynamics, respectively.
I noticed this structure only after reading through the three sections, which was confusing to me.
Explaining the structure of the discussion at an appropriate place (at the end of Introduction or at the beginning of Section 3) may improve the clarity.

2: Most practitioners use nonlinear VAEs, but the paper provides a negative result in that the VAE objective can induce a larger intrinsic dimension and larger support of data distribution.
Is there any wisdom or tip from the theory discussed in the paper for practitioners?
Such information will strengthen the importance of the paper.
For example, the impact of the choice of latent dimensionality $r$ or a way to choose $r$ can help VAE users.

Question about gradient flow analysis: can we extend to stochastic gradient setup?
In the linear case, the expectation over latent variable $z$ in the VAE objective is analytic.
On the other hand, we need a Monte-Carlo approximation for typical nonlinear cases using the reparameterization trick, etc.

**Summary Of The Paper:**

This paper examines the conjecture given in (Dai & Wipf; 2019) on the support of distribution that VAE learns.
The contributions are listed as follows:
- For the linear case where the data is Gaussian with rank-degenerate covariance, and encoder/decoder are both linear, this paper proves that VAE captures the intrinsic dimension of data distribution correctly by analyzing the objective and its gradient-flow dynamics.
- For nonlinear cases, the paper shows a counterexample to the conjecture in (Dai & Wipf; 2019) where the support of VAE generators is a superset of that of data distribution.
- Numerical experiments are presented to support the theory.

**Summary Of The Review:**

The paper gives a sensible discussion, but a lack of practical aspect limits the broader importance.

---

> ### Author Response · Authors · 2021-11-13
> **Thank you for your suggestions!**
>
> Thank you for the review! We’ll use your suggestions to improve the paper for the final version. We appreciate your feedback and will address to weaknesses below.
>
> **Adding a note regarding organization**: We agree with your suggestion to add some signposting language so that the reader understands better that Section 3 summarizes our results without being completely rigorous and Sections 4-5 give a rigorous presentation of the theoretical results. We have added language to that effect in the end of the introduction in the updated submission.
>
> **Regarding the results for SGD**: in fact, the bigger conceptual challenge is moving from gradient flow to gradient descent (i.e. discretizing the step size). As the loss goes to $-\infty$, the step size would have to decay with some schedule (depending on the parameters of the problem) so as not to deviate from the gradient flow. This seems technical, but within the realm of our techniques, and is an interesting problem for future work.

---

### Official Review · Reviewer_KKdc · 2021-11-02

**Correctness:** 3
**Technical Novelty And Significance:** 2
**Empirical Novelty And Significance:** 3
**Recommendation:** 5
**Confidence:** 4

**Main Review:**

Strength
- The work tackles previous work (Dai & Wipf, 2019), which is very interesting.
- Theorems and proofs are formally provided.

Weakness
- The problem statement or the motivation is not properly provided.
- The paper seems to be revised, and written in formal way.
- Linear VAE is barely used. Instead of linear VAE discussion, it would be better to discuss more on the non-linear VAE case, which is more widely used.
- No analysis in benchmark or real-world dataset. The authors only deal with toy datasets.

**Summary Of The Paper:**

The authors study further on the conjectures of Dai & Wipf (2019). For the linear case, they provide the proof that the conjecture is true. For the non-linear case, the paper disagrees with the conjecture, and they argue that the VAE training frequently learns a higher-dimensional manifold which is a superset of the ground truth manifold.


**Summary Of The Review:**

I'm negative on this paper. The paper needs to be revised in formal way. The motivation should be clearly introduced (rather than citing the paper), and the conclusion should be also provided. The paper seems not ready to be published, however, the work is interesting and I'm looking forward to have the paper (with additional experiment on benchmark or real-world) in the revised version.

---

> ### Author Response · Authors · 2021-11-13
> **Thank you for your suggestions**
>
> Thank you for your suggestions, we hope to use them to improve the paper further! We’ll respond to your points roughly in order below.
>
> **Problem statement, motivation**: Our high level motivation is to understand the behavior of VAEs in the case of data lying on a low-dimensional manifold. (E.g. from the introduction: “we study a common setting of interest where the data is supported on a low-dimensional manifold — often argued to be the setting relevant to real-world image and text data due to the manifold hypothesis.” We have statements of a similar nature in the abstract and setup.) We do acknowledge there are a large number of citations to Dai-Wipf in the paper, since we are directly studying the viability of an algorithm they proposed for this setting.
>
> **Level of formality in the paper**: We believe that we have formally and thoroughly stated our theorems, each one supported by a complete proof. Please let us know any concrete suggestions and we would be happy to make edits.
>
> **Results on benchmark data**:  Our work focuses on examining when VAEs are capable of recovering the support of a low dimensional distribution. In a real-world setting like MNIST or ImageNet, we do not have a notion of a ground truth manifold, nor an easy way to measure the dimension of the recovered manifold or the quality of recovery. It is for this reason that we work with synthetic data.
>
> **More results on nonlinear VAEs**: We’re happy to run additional experiments that the reviewer would like to see on the nonlinear case. Besides showing the existence of bad global optima, we do not prove any other theoretical results for the nonlinear case, as the simulations already show that the correct manifold would not be recovered — so there is nothing left to prove.

---

> > ### Comment · Reviewer_KKdc · 2021-11-29
> > **Thank you for your clarification**
> >
> > **Problem statement, motivation & the Level of formaility:** I understand that the paper relies on the previous work (Dai & Wipf, 2019), however, it would be better to start the paper by re-stating the previous work, and then go to the next step. I know that there is a paragraph which introduces the work of Dai & Wipf (2019), but how about re-writing the previous work in the authors' language, and then establishing the proposed work? While I do respect the work, but the paper is written in unkind shape that it was very hard to follow the current version of manuscript as a person who had to go all over the original work.
> >
> > **Results on benchmark data, More results on nonlinear VAEs:** Thank you for your clarification.
> >
> > Best

---

### Official Review · Reviewer_ayh8 · 2021-11-03

**Correctness:** 3
**Technical Novelty And Significance:** 2
**Empirical Novelty And Significance:** 2
**Recommendation:** 5
**Confidence:** 3

**Main Review:**

I believe that a deeper insight into the training behavior of VAEs is a very important research area. The paper provides further insight into possible degenerate optima that can be obtained by VAEs, both linear and nonlinear, during training.

However, I have several concerns regarding the accuracy of the paper. This is my current understanding, and I am fully willing to change my score if I have misunderstood anything.

1. Though the authors heavily cite and build on Dai & Wipf's paper, they do not explore whether the two-stage VAE proposed by Dai & Wipf remedies any of the problems posed in this paper. Without a central theorem or a clear theme that ties together the provided theorems, the current work feels somewhat disjointed.

2. I take issue with the way this paper positions itself with respect to Dai & Wipf. Namely, it misinterprets the theoretical results provided in the cited paper.
In the abstract, it is implied that Dai and Wipf’s result is on the convergence properties of the VAE during training, and that the present work expands on these convergence properties. See:
“Recent work by Dai and Wipf (2019) suggests that on low-dimensional data, the generator will converge to a solution with 0 variance which is correctly supported on the ground truth manifold. In this paper, via a combination of theoretical and empirical results, we show that the story is more subtle. Precisely, we show that for linear encoders/decoders, the story is mostly true and VAE training does recover a generator with support equal to the ground truth manifold, but this is due to the implicit bias of gradient descent rather than merely the VAE loss itself.”
However, it is my understanding that all of Dai & Wipf’s results do not consider any convergence behavior in training. Instead, they only analyze *optimal* behavior of the VAE. They do construct sequences of encoder/decoder functions that converge to these optima. But these sequences are unrelated to the training behavior of the VAE. Moreover, Dai & Wipf’s results only imply the generator only converges to a solution with 0 variance if the variance of the decoder goes to zero.

3. The first bullet point in Lemma 1, a restatement of Theorems 4 and 5 in Dai & Wipf, does not seem correct. Translating the notation in Dai and Wipf to that of this paper, Theorem 4 in Dai & Wipf implies the existence of a sequence of $f_t, g_t, D_t, \gamma_t$ such that $\gamma_t \rightarrow 0$ and $L(f_t, g_t, D_t, \gamma_t) \rightarrow -\infty$. Lemma 1 appears to state the converse, i.e. all sequences $f_t, g_t, D_t, \gamma_t$ s.t. $L(f_t, g_t, D_t, \gamma_t) \rightarrow -\infty$ have that $\gamma \rightarrow 0$.
Moreover, in both parts of Lemma 1, $f_t, g_t, D_t$ are allowed to be any sequence such that $\mathcal{L} \rightarrow -\infty$, whereas in Dai & Wipf they are fixed w.r.t. $t$ and optimal w.r.t. the loss $\mathcal{L}$.


Style:

Page 6. "There exists $\tilde{A}_1, \tilde{A}_2, \tilde{B}$ s.t. for $\tilde{\epsilon}_t \rightarrow 0$ there exists $\tilde{D}_t$ s.t." This setup for Theorem 4 was difficult to parse

Page 8. $(\sigma(\langle a^∗, \tilde{x}_{:r} \rangle) − \tilde{x}_{r+1})^2$ ...


should $\tilde{x}_{r+1}$ be instead $\tilde{x}_{r+1:}$?


**Summary Of The Paper:**

This paper builds on a recent theoretical work by Dai & Wipf [1]. Dai & Wipf analyze the *optimal* behavior of VAEs, when applied to manifold-valued data. This work analyzes the *training* behavior of VAEs, when applied to manifold-valued data. Both consider the non-trivial case where the manifold’s intrinsic dimension is lower than the ambient dimension. Dai & Wipf considers VAEs with arbitrarily complex encoders and decoders. The present work considers VAEs with linear encoders and linear/single-hidden layer nonlinear decoders.

[1] Bin Dai and David Wipf. Diagnosing and enhancing vae models. arXiv preprint arXiv:1903.05789,
2019.

**Summary Of The Review:**

While the paper provides insight in a very exciting area of theoretical research in machine learning, I currently do not recommend acceptance due to inaccuracies and the lack of a cohesive thesis.

Pros:
+ Novel theoretical analysis of VAEs during training and at global optima
+ Theoretical observations are corroborated by empirical results

Cons:
- Theory only considers linear encoders and linear/1-hidden layer nonlinear decoders
- The convergence behavior during training is only provided for linear VAEs
- Work feels disjointed, writing could be clearer (e.g. Theorem 4 has very little discussion)

**AFTER REBUTTAL**

My understanding of the work has been improved by the rebuttal. However, I do not feel that the stance taken in the manuscript adequately represents the stance taken in the rebuttal. I would be more willing to recommend acceptance if the misunderstandings were properly addressed in the manuscript---however, the manuscript, as it stands, has not been edited in such a way. (See my reply to the rebuttal.) I still stand by most of my points. Here is the updated summary:

Pros:
+ Novel theoretical analysis of VAEs during training and at global optima
+ Theoretical observations are corroborated by empirical results

Cons:
- Theory only considers linear encoders and linear/1-hidden layer nonlinear decoders
- Work feels disjointed, writing could be clearer (e.g. Theorem 4 has very little discussion)

---

> ### Author Response · Authors · 2021-11-13
> **Thank you for the detailed feedback!**
>
> We thank you for your feedback, and hope the clarifications here help improve the readability of our paper and move your score.
>
> **Thesis of our paper**: In brief, the goal of our paper is to show that the behavior of VAEs (in the limit of small decoder variance, when trained on low-dimensional data) is much more complicated than was claimed in a recent paper by Dai-Wipf (who in turn suggested a two-stage training algorithm based on these claims). More precisely, we show that the set of optima could contain distributions supported on the wrong manifold and that training does find some solutions with the wrong support, except in the case of linear VAEs (due to an implicit bias of the gradient dynamics). Thus, stage 1 of the two-stage approach in Dai-Wipf cannot work.
>
> **Thesis of Dai-Wipf**: The theory in the Dai-Wipf paper indeed analyzes the structure of the optima of the VAE loss, but their suggested 2-stage VAE algorithm is inspired by and relies on their theoretical results. To quote from their paper: “In brief, the first stage just learns the manifold [which] provides a mapping to a lower dimensional intermediate representation with no degenerate dimensions ... The second (much smaller) stage then only needs to learn the correct probability measure on this intermediate representation, which is possible per the analysis from Section 2”. Thus, we don’t feel we misrepresented the results in Dai-Wipf. Our results show that the reason the 2-stage training works better cannot be explained by the theoretical results in their paper.
>
> **Does the two-stage approach of Dai-Wipf remedy the issues we point out**: Our results exactly show that stage 1 of their two-stage approach cannot work. The goal of stage 1 is to learn a distribution that has the right support, but not necessarily the right distribution on it. Our results show that this will not happen, i.e. the first stage is not guaranteed to learn the right support, except for the case of linear VAEs.
>
> **Theory only concerns linear/1-hidden layer nonlinear decoders**: This is a feature, not a bug of our paper! Note, the thesis of our paper is that the correct manifold will only be recovered in the linear case. In the nonlinear case, we show that even for simple manifolds (e.g. high dimensional spheres, or data generated by a 1-layer sigmoidal decoder) the correct manifold will not be recovered.
>
> **No theory for training dynamics for nonlinear VAEs**: Continuing from the above point, we do not prove any theoretical results for the nonlinear case, because the simulations already show that the correct manifold would not be recovered — so there is nothing to be proven.
>
> **Regarding Lemma 1/Theorems 4+5 in Dai-Wipf**: The statement is correct as stated and the proof is in Appendix B.1. It is also roughly what Theorems 4+5 in Dai-Wipf show, and we credit them because the Lemma follows from arguments used in the proofs of those results. (The implication of Theorem 4 in Dai-Wipf is that for any set of parameters with loss converging to $-\infty$, the decoder variance has to go to 0; the implication of Theorem 5 is that for any such set of parameters, the reconstruction error also has to go to 0. Hence, the direction of the implication is if a sequence of parameters have loss going to $-\infty$, then the decoder variance and reconstruction error has to go to 0.)
>
> **Minor Suggestion**: the notation for page 8 experiment section is correct. We have $x_{r+1} = \sigma(\langle a^*, z \rangle)$ and the rest of the vector $x_{r+2:}$ is a zero padding. Thus to measure how well the VAE learnt the support of the distribution, we evaluate $\left(\sigma\left(\langle a^∗, \tilde{x}_{:r} \rangle\right)-\tilde{x}_{r+1}\right)^2$.

---

> > ### Comment · Reviewer_ayh8 · 2021-11-23
> > **Thank you for your clarifications**
> >
> > **Theorem 4 in Dai & Wipf vs as stated in your paper**
> > Here is Theorem 4 in Dai and Wipf:
> >
> > Let  $\{ \theta_\gamma^*, \phi_\gamma^* \}$  denote an optimal $\mathcal{k}$-simple VAE solution (with $\mathcal{k} \geq r$) where the decoder variance $\gamma$ is fixed. Moreover, we assume that $\mu_{gt}$ is not a Gaussian distribution when $d = r$. Then for any $\gamma > 0$, there exists a $\gamma' < \gamma$ such that $\mathcal{L}(\theta_{\gamma'\}^*, \phi_{\gamma'}^*) < \mathcal{L}(\theta_{\gamma\}^*, \phi_{\gamma}^*)$.
> >
> > This statement implies that, if the decoder variance goes to $0$, there exists a sequence of VAE parameters whose loss goes to $0$. You state that, for any sequence of VAE parameters whose loss goes to $0$, the decoder variance goes to $0$. These statements appear to be different to me. Namely, they are converse statements of each other.
> >
> > **Thesis of your paper**
> >
> > Thank you for clarifying your thesis. As it is stated in the comments, it makes sense to me. I will update my score to reflect this. However, I will argue that these points were not clear, especially in the abstract of the paper, and so I still suggest edits to the manuscript proper, rather than just a clarification in the rebuttal.
> >
> > Namely, in your abstract (emphasis mine): "Recent work by Dai and Wipf (2019) suggests that... the generator will $\textbf{converge}$ to a solution with 0 variance." -- This is not true, as you aptly quoted Dai and Wipf: "The second (much smaller) stage then only needs to learn the correct probability measure on this intermediate representation, which is $\textbf{possible}$ per the analysis from Section 2." I have trouble here namely with the bolded terms, i.e., **converging** to the desired parameters vs the **possibility** of the desired parameters.
> >
> > This misunderstanding is furthered in the statement that "for linear encoders/decoders, this **story [by Dai and Wipf] is mostly true** and VAE training does recover a generator with support equal to the ground truth manifold, but this is due to the **implicit bias of gradient descent**". Again, the implication is that the Dai and Wipf "story" is about the convergence properties of the VAE during training.
> >
> > While the difference here is subtle, it is crucial. And especially so in this paper, where the stated express goal in the paper is to shed further light on the subtleties of the arguments in Dai & Wipf.

---

> > > ### Author Response · Authors · 2021-11-23
> > > **Re: Thank you for your clarifications**
> > >
> > > Thank you for your response! We are happy to make edits to the paper—we were just not sure what the source of confusion was. Your latest reply helped narrow it down, and we hope further discussion will let us resolve it completely.
> > >
> > > **Re: Theorem 4 from Dai-Wipf**: You wrote: “This statement implies that, if the decoder variance goes to 0, there exists a sequence of VAE parameters whose loss goes to 0”. This is the *converse* of what Theorem 4 from Dai-Wipf you copied says. The theorem says, for any value of $\gamma > 0$, we can find a smaller $\gamma’$ s.t. the optimal model parameters corresponding to this $\gamma’$ give a smaller loss than the optimal parameters corresponding to $\gamma$. The conclusion is that *if a sequence of parameters $(\gamma_t, \theta_t, \phi_t)$ converges to loss $-\infty$, it must be the case that $\gamma_t \to 0$.*
> > >
> > > **Re: the subtleties in the abstract**: Do we understand correctly that your point of contention is that Dai-Wipf don’t *prove* that the gradient descent dynamics will converge to the desired optima of the VAE loss? If so, we are happy to add some language to this effect in the abstract. We completely agree with this—in fact, our paper shows this is *not* the case.
> > >
> > > If, however, the gradient descent dynamics *do not* converge to one of the desired optima, the two-stage algorithm proposed in their paper does not make sense. (In other words, the algorithm is predicated upon this happening.) Regarding your bolded words: “possible” in the quote we included does not mean “it is possible that stage 1 recovers the correct optima”; rather, it means “it is possible to view stage 2 as learning a distribution over the correct manifold”. Note that the referenced quote says that “the second (much smaller) stage then **only needs** to learn the correct probability measure on this intermediate representation” which only makes sense if stage 1 is learning the manifold successfully.
> > >
> > > Moreover, if the point of contention is that we are attributing claims that are too strong to Dai-Wipf, we are happy to try to tone them down. We used phrases like “Dai-Wipf suggests”, and “Dai-Wipf conjecture” to indicate that there were no end-to-end formal mathematical claims about their algorithm.

---

> > > > ### Comment · Reviewer_ayh8 · 2021-11-27
> > > > **Thank you for the quick response**
> > > >
> > > > **Re: Re: Theorem 4 from Dai and Wipf**
> > > > This is not true. Theorem 4 implies that, given a set of optimal parameters $\phi$, $\theta$, one can generate a sequence of parameters that diverge in loss to $-\infty$, by taking $\gamma \rightarrow 0$. Call this implied result Corollary A. However, Theorem 4 does not imply your statement that all sequences $\phi_i, \theta_i$ that diverge to $-\infty$ require that $\gamma \rightarrow 0$. Call your statement Statement B. Of course, Statement B is true: if you inspect the loss $\mathcal{L}$, the term involving $\gamma$ is the only one unbounded below, and so it must go to $0$ when $\mathcal{L} \rightarrow -\infty$. But this observation does not require Theorem 4. Therefore, while your statement is indeed correct, I do not see its logical relationship to Theorem 4.
> > > >
> > > > That being said, it appears to me that Statement B does imply Corollary A. And all three results are quite apparent from the same above-mentioned observation.
> > > >
> > > > **Re: Re: the subtleties in the abstract**
> > > > Yes, precisely. I believe that the language in your paper misrepresents Dai and Wipf, and their analysis of VAE optima. These details are crucial for a paper that aims to build on the prior paper. Furthermore, I would argue that some of the relevance of your analysis comes from the way it positions itself with respect to Dai and Wipf. Therefore, you may have given other reviewers the wrong impression. Thus, rather than making such changes after the submission deadline (and also the revision deadline), I would lightly recommend resubmission.

---

> > > > > ### Author Response · Authors · 2021-11-28
> > > > > **Re: Thank you for the quick response**
> > > > >
> > > > > Thanks for your reply.
> > > > >
> > > > > **Re: Theorem 4 in Dai-Wipf**: We both agree that “Statement B” is true and the disagreement seems to be what statement Theorem 4 in Dai-Wipf corresponds to.
> > > > >
> > > > > You wrote: “Theorem 4 implies that, given a set of optimal parameters $\phi$, $\theta$, one can generate a sequence of parameters that diverge in loss to $-\infty$, by taking $\gamma \rightarrow 0$. Call this implied result Corollary A.” This fact, although true, *does not* follow from the statement of Theorem 4 in Dai-Wipf. Theorem 4 says that for any decoder variance $\gamma$, there exists $\gamma’ < \gamma$ such that a smaller loss is achieved for the optimal VAE parameters with decoder variance $\gamma’$. It does not imply that the loss will be driven to $-\infty$ by some sequence of VAEs, because the loss could just as well asymptote to 0 (e.g., optimal VAE parameters with $\gamma = 0.1$ gives loss $0.05$, with $\gamma = 0.01$ gives loss $0.01$, ...) or any other number instead of $-\infty$ and this would be consistent with the Theorem statement. On the other hand, it does imply: “If $\gamma$ is not constrained, it must be that $\gamma \to 0$ if we wish to minimize [the objective]” as is in fact written right below Theorem 4 in Dai-Wipf. This is because if we look at the best performing VAE with $\gamma = 0.1$, for example, it cannot be the unconstrained global optimum as it performs strictly worse than some VAE parameters with a smaller value of $\gamma$. This is why above Theorem 4, Dai-Wipf describe the result as being about “some necessary conditions for VAE optima”. We hope this clarifies why Theorem 4 in Dai-Wipf corresponds to Statement B and not Corollary A in your terminology, and that neither of those statements directly imply each other.
> > > > >
> > > > > **Re: subtleties in abstract:** We received your reply to our rebuttal after the update deadline, so there was no chance for us to modify anything in the submission. As a sign of good faith on our part, here are some suggested changes that we believe address your concerns:
> > > > >
> > > > > *New abstract:*
> > > > >
> > > > > Variational Autoencoders (VAEs) are one of the most commonly used generative models, particularly for image data.  A prominent difficulty in training VAEs is data that is supported on a lower dimensional manifold. Recent work by Dai and Wipf (2020) proposes a two-stage training algorithm for VAEs, based on a conjecture that in standard VAE training the generator will converge to a solution with 0 variance which is correctly supported on the ground truth manifold. They gave partial support for this conjecture by showing that *some* optima of the VAE loss do satisfy this property, but did not analyze the training dynamics. In this paper, we show that for linear encoders/decoders, the conjecture is true—that is the VAE training does recover a generator with support equal to the ground truth manifold—and does so due to an implicit bias of gradient descent rather than merely the VAE loss itself. In the nonlinear case, we show that VAE training frequently learns a higher-dimensional manifold which is a superset of the ground truth manifold.
> > > > >
> > > > > *Change to intro:*
> > > > >
> > > > > We suggest changing the following paragraph:
> > > > >
> > > > > “In this setting, Dai & Wipf (2019) proposed a two-stage training process for VAEs, based on a combination of empirical and theoretical arguments suggesting that for standard VAE training with such data distributions: (1) the generator’s covariance will converge to 0, (2) the generator will learn the correct manifold, but not the correct density on the manifold (3) the number of approximately 0 eigenvalues in the encoder covariance will equal the intrinsic dimensionality of the manifold (see also Dai et al. (2017)).”
> > > > >
> > > > > to the following paragraph:
> > > > >
> > > > > “In this setting, Dai & Wipf (2019) proposed a two-stage training process for VAEs, based on a conjecture that for standard VAE training with such data distributions: (1) the generator’s covariance will converge to 0, (2) the generator will learn the correct manifold, but not the correct density on the manifold (3) the number of approximately 0 eigenvalues in the encoder covariance will equal the intrinsic dimensionality of the manifold (see also Dai et al. (2017)). Formally, they showed that some optima of the VAE loss satisfy this conjecture, but they did not attempt to analyze the training dynamics.”

---

> ### Author Response · Authors · 2021-11-22
> **Have we addressed your concerns?**
>
> We thank the reviewer again for their detailed review. As the review period is drawing to a close, we wish to followup and check whether our response addressed the points of concern and confusion. Please let us know if we there is anything else we can clarify!

---

### Decision · Program_Chairs · 2022-01-20

**Decision:**

Accept (Poster)

**Comment:**

The paper analyzes the behavior of VAEs in modeling data lying on a low dimensional manifold. It formally proves some of the conjectures/informal-statements in an earlier work by Dai and Wipf (2019) in the case of linear VAE and linear manifold, and disproves the same for the nonlinear case. In particular, it proves, by analyzing the objective and its gradient-flow dynamics, that VAE captures the intrinsic dimension of data distribution correctly. For nonlinear cases, the paper shows a counterexample to the conjecture in (Dai & Wipf; 2019) where the support of VAE generators is a superset of that of data distribution.

Two of the reviewers had raised following specific concerns -- (i) Theory only considers linear encoders and linear/1-hidden layer nonlinear decoders, (ii) The convergence behavior during training is only provided for linear VAEs, (iii) Some statements in the introduction/abstract misrepresent the results in (Dai & Wipf; 2019). However the authors have adequately addressed (i) and (ii) in their response -- the paper shows that the correct manifold will only be recovered in the linear case; in the nonlinear case, even for simple manifolds (1-hidden layer) the correct manifold is not recovered as shown by the counterexamples. Authors have also promised to modify the statements in the abstract and introduction to address the concern in (iii). Other two reviewers are largely positive about the paper. The paper makes an important contribution to the VAE literature in further clarifying VAEs' behavior while modeling low dimensional manifolds, and will be a good addition to the conference program.